# FlowNet: Modeling Dynamic Spatio-Temporal Systems via Flow Propagation

**Yutong Feng[1], Xu Liu[2], Yutong Xia[2], Yuxuan Liang[1]***

[1]The Hong Kong University of Science and Technology (Guangzhou)
[2]National University of Singapore
ytfeng.caspian@163.com;liuxu726@gmail.com
{yutong.x,yuxliang}@outlook.com

## Abstract

Accurately modeling complex dynamic spatio-temporal systems requires capturing flow-mediated interdependencies and context-sensitive interaction dynamics. Existing methods, predominantly graph-based or attention-driven, rely on similarity-driven connectivity assumptions, neglecting asymmetric flow exchanges that govern system evolution. We propose Spatio-Temporal Flow, a physics-inspired paradigm that explicitly models dynamic node couplings through quantifiable flow transfers governed by conservation principles. Building on this, we design FlowNet, a novel architecture leveraging flow tokens as information carriers to simulate source-to-destination transfers via Flow Allocation Modules, ensuring state redistribution aligns with conservation laws. FlowNet dynamically adjusts the interaction radius through an Adaptive Spatial Masking module, suppressing irrelevant noise while enabling context-aware propagation. A cascaded architecture enhances scalability and nonlinear representation capacity. Experiments demonstrate that FlowNet significantly outperforms existing state-of-the-art approaches on seven metrics in the modeling of three real-world systems, validating its efficiency and physical interpretability. We establish a principled methodology for modeling complex systems through spatio-temporal flow interactions.

## 1 Introduction

Accurately modeling the evolution of complex dynamic spatio-temporal systems remains a fundamental challenge in domains ranging from urban mobility planning [1, 2] to environmental monitoring [3]. We identify a critical yet understudied paradigm governing such systems: the mutual interactions between distributed entities through directed information flow exchanges [4, 5]. These flow-driven dynamics not only modulate the system states [6] but also serve as primary catalysts for emergent spatio-temporal patterns [7, 8].

For decades, extensive research has focused on effectively predicting spatio-temporal systems [9, 10] and has demonstrated remarkable success by jointly modeling spatial dependencies and temporal dynamics [11, 12]. Graph-based architectures, particularly spatio-temporal graph neural networks (STGNNs) [13, 14, 15], leverage structural priors by constructing adjacency matrices from predefined spatial relationships like geographic proximity or semantic connections. Attention-based methods, especially Transformers [16, 17, 18, 19], introduce complementary advantages through dynamically computed node affinity matrices, enabling adaptive modeling of non-stationary dependencies while maintaining favorable scalability. Current approaches further bridge this framework with the foundation models [20, 11], enabling the extraction of spatio-temporal dependencies from various data streams [21, 22, 23, 24].

---

*Y. Liang is the corresponding author. Email: yuxliang@outlook.com

39th Conference on Neural Information Processing Systems (NeurIPS 2025).

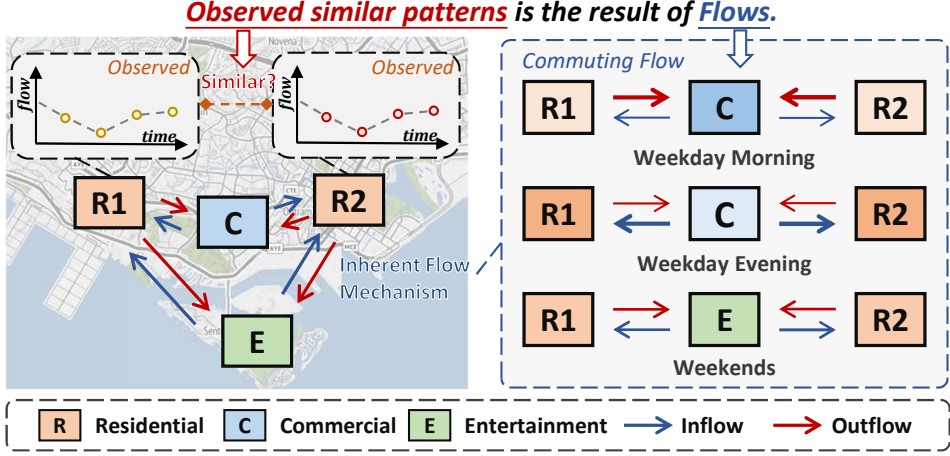

Figure 1: An illustration of the flow mechanism (case in an urban system). Residents move between zones at different times, thus presenting variations in flow observations in each zone. Specifically, darker colors and thicker arrows indicate larger values. Similar observations are presented between different residential zones (R1 & R2), but the similarity itself is only the symptom of the system, not the intrinsic evolutionary drive.

Although effective, existing approaches remain constrained by a critical oversight: they primarily capture surface-level observational correlations while fundamentally neglecting the flow-mediated interdependencies that govern intrinsic system dynamics. Specifically, prevalent operators like graph convolution [25, 26] and spatial attention [27, 28, 16, 19] implicitly assume system dynamics emerge from static node attribute similarities, whereas a range of real-world interconnected systems evolve through asymmetric flow exchanges that transcend mere statistical correlation [29, 30, 31, 32]. Consider an urban system comprising residential and commercial zones as shown in Figure 1, where directional population movement creates distinct spatio-temporal patterns: morning peaks exhibit concentrated flows from residential to commercial areas, while evening peaks reverse this directional pattern. Crucially, while spatially distant residential zones may exhibit similar traffic volumes through observational metrics, the underlying system dynamics derive from directional flow interactions between functionally complementary regions. The critical modeling imperative lies in capturing how transit flows actively reconfigure system states across different time, rather than correlating static attribute similarities among different zones.

Furthermore, the inherent complexity of modeling spatio-temporal systems arises from the context-sensitive dynamics governing flow propagation [33, 34]. These dynamics exhibit dual heterogeneity: *temporal* variations in flow magnitude and directionality during peak hours alongside *spatial* shifts in functional destinations during holidays. As previously discussed, the flows that form the morning and evening peaks within the city show an opposite direction. During holiday periods, leisure-oriented mobility redirects flows toward entertainment venues instead of workplaces. Conventional approaches attempt to resolve this complexity through two restrictive strategies. Predefined connectivity graphs [25, 26] enforce static spatial priors that cannot adapt to contextually evolving interaction ranges. Conversely, pairwise similarity computations [35, 36, 16, 19] dynamically estimate node affinities but inherently aggregate noise from irrelevant connections.

The intrinsic limitations of existing methodologies necessitate a paradigm shift in modeling dynamic spatio-temporal systems. Observing that transportation networks, urban mobility, or hydrological systems share the common intrinsic structure of latent information flow propagation, we propose a new paradigm termed **Spatio-Temporal Flow**. This physically inspired framework departs from traditional approaches constrained by static or similarity-based connectivity assumptions by explicitly modeling dynamic node coupling through mobile information carriers. Additionally, information transfer adheres to explicit conservation laws where outflow operations deplete source node states while inflow operations proportionally augment destination states. Finally, interaction ranges adaptively adjust based on contextual system states, dynamically reconfiguring propagation pathways without relying on fixed thresholds. This framework fundamentally contrasts with previous schemes that propagate information through merely blending node features while ignoring the state redistribution mechanisms inherent to flow-mediated systems.

**Contribution.** Building upon this paradigm, we propose **FlowNet**, a novel prediction model based on spatio-temporal flow. Unlike conventional message passing through feature blending (e.g., attention or graph convolution), FlowNet introduces **flow tokens** as quantifiable information carriers that explicitly model source-to-destination transfers. These tokens are generated and redistributed between nodes through **Flow Allocation Modules (FAM)**, ensuring compliance with the conservation principle. To address the dynamic propagation ranges across spatio-temporal contexts, FlowNet utilizes an **Adaptive Spatial Masking (ASM)** module that learns a node-specific adaptive interaction radius where information exchange occurs only between nodes within this radius, eliminating noise from irrelevant distant nodes. Furthermore, we design a cascaded architecture with hyper-connection [37] and Mixed Multi-Layer Perceptron (M-MLP), enabling dynamic adaptation of inter-layer connectivity strengths and branch interactions. Our experiments demonstrate that FlowNet achieves statistically significant improvements over existing state-of-the-art approaches across seven key metrics when modeling three challenging real-world systems.

## 2 Similarity vs. Intrinsic Flow: Which is better?

In this section, we formalize the modelling of the dynamic spatio-temporal systems and the key concepts underlying our proposed paradigm. Let a spatio-temporal system be represented as a graph $\mathcal{G} = (\mathcal{V}, \mathcal{E})$, where $\mathcal{V} = \{v_1, \ldots, v_N\}$ denotes $N$ nodes such as traffic sensors or geographic regions, and $\mathcal{E}$ defines edges. The system's state at time $t$ is characterized by node observations $\mathbf{X}^t \in \mathbb{R}^N$, which capture measurable attributes such as traffic speed or water level. Given historical observations $\{\mathbf{X}^{t-T}, \ldots, \mathbf{X}^{t-1}\}$, the task aims to predict future states $\{\mathbf{X}^t, \ldots, \mathbf{X}^{t+\tau}\}$.

Traditional similarity-driven methods, such as graph convolution and spatial attention, fundamentally operate by blending node features based on predefined or dynamically inferred similarity metrics. While these approaches effectively capture surface-level correlations in observational data, they conflate statistical associations with the intrinsic mechanisms driving system evolution. By prioritizing feature proximity over directional flow dynamics, these methods fail to disentangle observational correlations from the flow-mediated interdependencies that actively redistribute system states. In contrast, our flow-based framework adopts a first-principles perspective, modeling system dynamics through intrinsic flow interactions. Specifically, we propose three guidelines for the framework:

- **Flow-Centric System Dynamics**: All interactions between nodes across varying spatio-temporal contexts are mediated by flow tokens $\phi \in \mathbb{R}^d$, which serve as quantifiable information carriers. A node's state is determined by its accumulated flow tokens, and system evolution emerges from the directed redistribution of $\phi$ across nodes and time steps.

- **Conservation Laws**: Information transfer between nodes follows a source-destination conservation law. If $v_j$ transmits a flow token $\phi^t_{j \to i}$ to $v_i$, the source token $\Phi_j$ depletes by $\phi^t_{j \to i}$ while the destination token $\Phi_i$ increases proportionally.

- **Adaptive Propagation Domain**: For each node $v_i$, its interaction neighborhood $\mathcal{N}^t_i \subseteq \mathcal{V}$ at time $t$ is determined by a learnable, context-aware radius $r^t_i$, discarding interactions beyond $r^t_i$.

## 3 Methodology

We present **FlowNet**, a physics-inspired architecture that re-formulates the modeling of spatio-temporal dynamics through flow-based information propagation, as shown in Figure 2. FlowNet first transforms the system state into latent representations via a patchify module, then progressively processes these features through cascaded modules to capture the spatio-temporal evolution of the system, and ultimately decodes the refined patterns through a projection layer to generate future state predictions. Our design dynamically constrains node interactions via Adaptive Spatial Masking (ASM), which prevents non-physical long-range dependencies by restricting communication to physically plausible regions. Central to the FlowNet are learnable flow tokens $\phi \in \mathbb{R}^d$ that serve as adaptive carriers for spatio-temporal information exchange. Specifically, FlowNet first embeds each node's observation into a high-dimensional flow space $\mathbb{R}^d$ through patching. Each node generates and distributes its own flow tokens through the Flow Allocation Modules (FAMs). Based on these, different FAMs are cascaded with each other via hyper-connection and M-MLP to enhance the nonlinearity of the model.

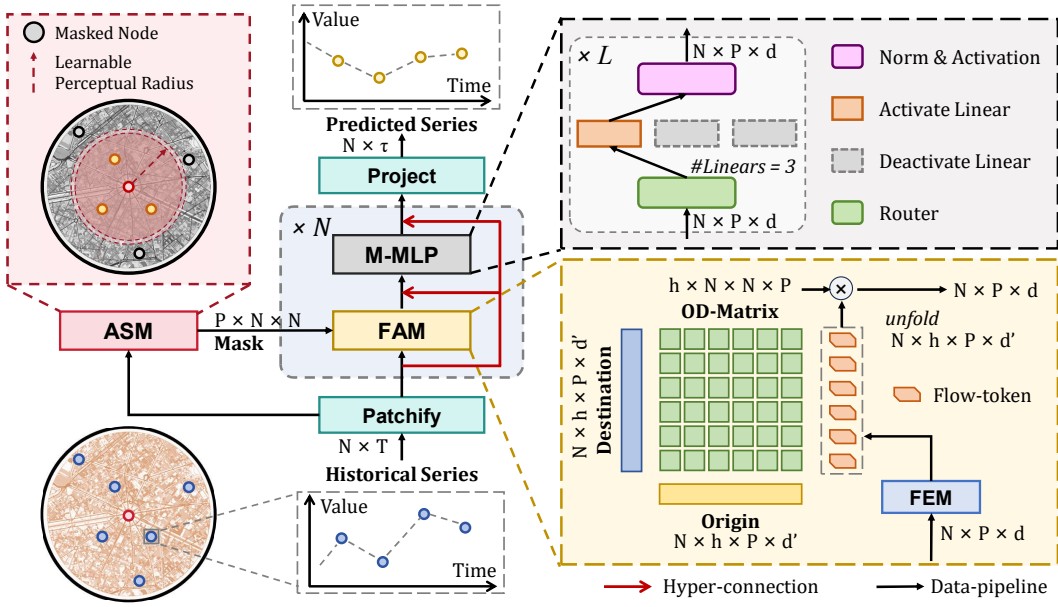

Figure 2: The overall structure of FlowNet, which consists of: (i) **Adaptive Spatial Masking (ASM)** dynamically adjusts interaction radius to filter irrelevant nodes; (ii) **Flow Allocation Modules (FAMs)** enforce source-to-destination transfer with explicit conservation of information mass; (iii) a **cascaded architecture** with **hyper-connections** and **Mixed Multi-Layer Perceptron (M-MLP)** further enhances multi-scale representation learning.

## 3.1 Preprocess

We initially construct the latent state of the spatio-temporal system by structured sequence tokenization, a process designed to capture temporal patterns. Given an input sequence $\mathbf{X} \in \mathbb{R}^{N \times T}$ representing $N$ nodes over $T$ historical timesteps, we first partition each node's univariate series into $P$ localized patches via a sliding window operation:

$$\mathbf{X}'_p = \text{Partition}(\mathbf{X}; P, S) \in \mathbb{R}^{N \times P \times M}, \tag{1}$$

where $M$ denotes the patch length and $S$ is the stride size controlling overlap between adjacent patches ($S < M$ for overlapping regimes). Each patch is then projected into the flow space via a parameterized embedding:

$$\mathbf{X}_p = \mathbf{X}'_p \mathbf{W}_e + \mathbf{b}_e \in \mathbb{R}^{N \times M \times d}, \tag{2}$$

where $\mathbf{W}_e \in \mathbb{R}^{P \times d}$ and $\mathbf{b}_e \in \mathbb{R}^d$ are learnable parameters. The patching process reduces the number of input tokens to the model by a factor of $S$ and significantly increases the model runtime performance in comparison with standard point-wise approaches. For convenience, we utilize the subscript $t$ to refer to different patches in the next subsections.

## 3.2 Adaptive Spatial Masking

Spatio-temporal systems inherently follow Tobler's First Law of Geography [38] - nearby locations tend to share similar characteristics through spatial interactions. However, real-world observations reveal that the effective interaction range between locations dynamically changes across space and time. We propose the Adaptive Spatial Masking (ASM) mechanism to address this gap by enabling each node to automatically determine its context-aware interaction range, adapting to both geographical constraints and transient system states. For each node $v_i$, we first augment its temporal features $\mathbf{X}'_i \in \mathbb{R}^{P \times d}$ with a learnable node-specific embedding $\mathbf{E}_i \in \mathbb{R}^d$, capturing static attributes (e.g., geographic coordinates or infrastructure properties):

$$\tilde{\mathbf{X}}_i = \text{Concat}\left(\mathbf{X}'_i, \mathbf{E}_i \odot \mathbf{1}_P\right) \in \mathbb{R}^{P \times 2d}, \tag{3}$$

where $\odot$ denotes broadcasting and $\mathbf{1}_P$ is a ones vector replicating $\mathbf{E}_i$ across all $P$ patches for aligning the dimension. The perception radius $r_i^t \in \mathbb{R}^+$ for node $v_i$ at patch $t$ is dynamically predicted via:

$$r_i^t = \texttt{Softplus}\left(\tilde{\mathbf{X}}_i^t \mathbf{W}_h + \mathbf{b}_e\right), \tag{4}$$

where $\mathbf{W}_h \in \mathbb{R}^{2d \times 1}$ and $\mathbf{b}_e \in \mathbb{R}^{1 \times 1}$ are learnable weights. The $\texttt{Softplus}$ operation ensures a non-negative radius while maintaining gradient stability. A time-varying spatial mask $\mathbf{M}_t \in \mathbb{R}^{N \times N}$ is then constructed based on geographic distances $d_{ij}$ between node $v_i$ and $v_j$:

$$\mathbf{M}_t[i,j] = \texttt{Sigmoid}\left(r_i^t - d_{ij}\right). \tag{5}$$

We use a $\texttt{Sigmoid}$ function to make the mask a differentiable operation, which in turn can be optimized by gradient descent and backpropagation during training. The measure of $d_{ij}$ can be determined according to the specific spatio-temporal system. For instance, the Manhattan distance can be chosen to measure the distance between nodes in an urban context, while the Euclidean distance can be utilized in a spatially isotropic natural system. By choosing different distance measurements and introducing the learnable perceptual radius, ASM is able to dynamically decide the propagation range of the flow tokens of each node based on the spatio-temporal context, thus filtering out non-physical long-distance dependent interference.

### 3.3 Flow Allocation Modules

To model spatio-temporal dependencies arising from flow token exchange between nodes, we propose Flow Allocation Modules (FAMs). At their core, FAM incorporates two Flow Estimation Modules (FEMs) with identical architectures but distinct learnable parameters. One FEM estimates the initial tokens $\mathbf{\Phi}_o$ retained by nodes, while the other predicts the allocated tokens $\mathbf{\Phi}_a$ distributed by nodes.

Notably, while standard attention mechanisms show limitations in capturing spatial dependencies through our analysis, they remain effective for temporal modeling when applied to local temporal patches. Building on this insight, we implement each FEM using Causal Temporal Multi-head Self-Attention (CTMSA) [3] to process time-ordered patch sequences. Formally, given augmented node features $\tilde{\mathbf{X}} = [\tilde{\mathbf{X}}_1, \cdots, \tilde{\mathbf{X}}_N] \in \mathbb{R}^{N \times P \times d}$, CTMSA produces:

$$\text{FEM}: \quad \hat{\mathbf{\Phi}} = \texttt{CTMSA}\left(\tilde{\mathbf{X}}\mathbf{W}_f + \mathbf{b}_f\right) \in \mathbb{R}^{N \times h \times P \times d'}, \tag{6}$$

where $\mathbf{W}_f, \mathbf{b}_f$ are learnable parameters of affine transformation, $h$ denotes attention heads and $d' = d/h$. Here, $\hat{\mathbf{\Phi}}$ corresponds to the head-folded representations of $\mathbf{\Phi}_o$ and $\mathbf{\Phi}_a$, and we utilize this superscript to denote the tensor reshaped into multiple heads in the subsequent part. The FEMs operate in a channel-independent manner, while we retain the vanilla multi-head design to enhance the characterization of token.

Given the estimated tokens $\mathbf{\Phi}_o$ (retained) and $\mathbf{\Phi}_a$ (allocated) from the FEMs, we next compute the flow distribution between neighboring nodes. For node $v_i$ at patch $t$, we first transform its augmented features $\tilde{\mathbf{X}}_i^t$ through an affine projection followed by node-wise layer normalization, yielding $\dot{\mathbf{X}}_i^t \in \mathbb{R}^d$. From this, we derive two dynamic representations via linear transformations:

- Origin vector $\mathbf{O}_i^t \in \mathbb{R}^d$ encoding $v_i$'s role as a token source.
- Destination vector $\mathbf{D}_i^t \in \mathbb{R}^d$ encoding $v_i$'s role as a token recipient.

These vectors are augmented with static node properties through learnable head-folded embeddings $\mathbf{E}_{i,O}, \mathbf{E}_{i,D} \in \mathbb{R}^{d'}$, yielding multi-head representations $\hat{\mathbf{O}}_i^t$ and $\hat{\mathbf{D}}_i^t$. The flow logit from $v_i$ to neighbor $v_j \in \mathcal{N}_i^t$ is then computed as follows:

$$q_{ij}^t = \alpha \cdot (\hat{\mathbf{O}}_i^t)^\top \hat{\mathbf{D}}_j^t \cdot \mathbf{M}_t[i,j], \tag{7}$$

where $\alpha = 1/\sqrt{d'}$ is the scaling factor that stabilizes gradient magnitudes, and $\mathbf{M}_t[i,j]$ is the element in the spatial mask learned by ASM. The normalized transfer probability is obtained via:

$$p_{ij}^t = \frac{\exp(q_{ij}^t)}{\sum_{k \in \mathcal{N}_i^t} \exp(q_{ik}^t)}. \tag{8}$$

Thus, the final tokens $\hat{\phi}_i^t \in \mathbb{R}^{h \times d'}$ that $v_i$ owns can be formulated as:

$$\hat{\phi}_i^t = \hat{\phi}_{i,o}^t - \hat{\phi}_{i,a}^t + \sum_{i \subseteq \mathcal{N}_k^t} \hat{\phi}_{k,a \to i}^t, \tag{9}$$

where $\hat{\phi}_{i,a}^t$ is the tokens sent out by $v_i$, and $\hat{\phi}_{k,a \to i}^t = p_{ki}^t \cdot \hat{\phi}_{k,a}^t$ is the tokens received by $v_i$. Subsequently, the matrix expression for the flow allocation process at $t$ can be formulated as:

$$\hat{\mathbf{\Phi}}^t = \hat{\mathbf{\Phi}}_o^t - \hat{\mathbf{\Phi}}_a^t + \mathbf{\Lambda}^t \hat{\mathbf{\Phi}}_a^t = \hat{\mathbf{\Phi}}_o^t + (\mathbf{\Lambda}^t - \mathbf{I})\hat{\mathbf{\Phi}}_a^t, \tag{10}$$

where $\hat{\mathbf{\Phi}}^t \in \mathbb{R}^{h \times N \times d'}$ is the flow for all nodes at $t$. $\mathbf{\Lambda}^t \in \mathbb{R}^{N \times N}$ is the allocation matrix at $t$ where $\mathbf{\Lambda}^t[i, j] = p_{ij}^t$. By separating retained and allocated tokens, FAM enforces source depletion and destination enhancement during transfers.

### 3.4 Cascading of Modules

We introduce hyper-connection [37] instead of normal residual connections to optimize the way modules are stacked. The hyper-connection introduces depth-connection to assign weights to the connections between the inputs and outputs of each module, and includes width-connection to allow information exchange within the same layer. Unlike rigid residual connections, these depth- and width-aware connections adaptively scale information flow, ensuring that multi-scale flow dynamics are hierarchically aggregated within deep network structures. Additionally, We intersperse MLP units between different FAM modules. After directional exchange via FAM, flow tokens pass through an MLP that fuses information across their multi-head representations. We replace vanilla linear layers with a Mixture of Linears (MoL). Unlike Mixture of Experts (MoE), which treats entire blocks as experts, MoL treats each linear projection as an independent expert. For an MLP with $L$ layers, this design creates a combinatorial parameter space of $L \times E$ distinct linear transformations compared to only $E$ transformations in equivalently sized MLP-MoE designs.

## 4 Experiments

In this section, we conduct experiments to evaluate FlowNet's performance and reveal its underlying mechanisms. Our experiments are designed around the following Research Questions (RQ):

- **RQ1**: How does FlowNet perform compared to existing Spatio-Temporal forecasting approaches?
- **RQ2**: How does each component in the flow dynamics contribute to improving model performance?
- **RQ3**: How efficient is FlowNet compared to other models?
- **RQ4**: What distribution do the node degree and perceptual radius learned through ASM obey?
- **RQ5**: What are the differences between the allocation matrix $\mathbf{\Lambda}$ learnt through FAM and the attention map learnt through the spatial attention-based methods?

### 4.1 Experimental Setups

**Datasets.** We evaluate our approach on three datasets (PEMS04F [36], DeepBase [39], SINPA [40]) of dynamic spatio-temporal systems, spanning transportation, hydrology, and urban mobility domains. To compare the forecasting effectiveness of FlowNet with other baselines on different temporal forecasting scales, we set up both short-term forecasting and long-term forecasting tasks on each dataset. The historical and predicted step sizes utilised were aligned across all tasks. Further details are listed in Appendix B.

**Baselines.** We include ten state-of-the-art methods for forecasting performance comparisons, including Autoformer [41], PatchTST [42], Crossformer [43], iTransformer [44], FEDformer [45], STGCN [25], GWNET [35], SCINet [46], STTN [16], and STAEformer [19].

**Implementation Details.** We conduct all experiments on one NVIDIA A100 80GB GPU. The Adam optimizer is utilized to train our model, and the batch size is 8. The learning rate starts from $1 \times 10^{-3}$, halved every 20 epochs until the $60_{th}$ epoch, and we start early stopping at the $20_{th}$ epoch of training. For FlowNet, we stack 2 layers of FAM for FlowNet and set up 16 experts inside the M-MLP. the Flow token uses 4 heads, and all hidden layer dimensions are set to 64. We leverage Mean Absolute Error (MAE) and Root Mean Squared Error (RMSE) for evaluation, where a smaller metric means better performance. More details can be found in Appendix B.

## 4.2 Model Comparison (RQ1)

In this section, we perform a model comparison in terms of MAE and RMSE. We run each method five times with different random seeds and report the average metric of each model. According to the results in Table 1, our proposed FlowNet model demonstrates remarkable improvements over the baseline models across all datasets and metrics. This outcome validates the effectiveness of our model in handling spatio-temporal data derived from diverse domains and varying spatial scales. Notice that FlowNet has taken the lead on both long-term and short-term prediction tasks, whereas none of the previous methods (STGNN or Transformer-based models) have been able to maintain a consistent advantage on a particular dataset or task. We identify that there are two potential reasons for FlowNet to perform well. Firstly, the flow mechanism is a more intrinsic modeling of the system compared to similarity capture, allowing FlowNet to accurately predict the short- and long-term evolution of the system. Secondly, the deployment of ASM allows FlowNet to dynamically determine the propagation range of the flow based on the spatio-temporal context, which allows it to be able to adapt to diverse spatio-temporal systems.

Table 1: Model performance comparison. The **bold**/underlined font means the best/second-best result. 'OOM' means that the model incurs out-of-memory issues on an A100 80GB GPU. * denotes the improvement of FlowNet over the second-best model is statistically significant at level 0.05.

| Dataset | | PEMS04F | | | | DeepBase | | | | SINPA | | | |
|---|---|---|---|---|---|---|---|---|---|---|---|---|---|
| Task | Short-term | | Long-term | | Short-term | | Long-term | | Short-term | | Long-term | | |
| Metrics | MAE | RMSE | MAE | RMSE | MAE | RMSE | MAE | RMSE | MAE | RMSE | MAE | RMSE |
| Autoformer | 35.29 | 51.68 | 107.67 | 136.10 | 0.79 | 1.47 | 0.84 | 1.69 | 122.45 | 215.66 | 131.43 | 234.16 |
| PatchTST | 26.82 | 41.01 | 29.85 | 48.38 | 0.59 | 1.21 | 0.62 | 1.34 | 68.40 | 115.87 | 40.44 | 69.36 |
| Crossformer | 19.14 | 29.70 | 26.15 | 43.95 | 0.58 | 1.19 | 0.69 | 1.42 | 64.13 | 108.24 | 62.49 | 97.76 |
| iTransformer | 21.52 | 32.73 | 28.58 | 44.25 | 0.61 | 1.27 | 0.70 | 1.45 | 84.53 | 143.69 | 108.32 | 199.54 |
| FEDformer | 20.53 | 31.55 | 58.49 | 80.37 | 0.80 | 1.46 | 0.88 | 1.66 | 110.44 | 198.48 | 109.77 | 205.70 |
| STGCN | 20.98 | 32.26 | 27.91 | 44.76 | 0.50 | 1.06 | 0.51 | 1.10 | 65.17 | 112.54 | 33.41 | 62.86 |
| GWNET | 18.82 | 29.36 | 26.38 | 42.10 | 0.49 | 1.02 | 0.66 | 1.38 | 59.04 | 103.75 | 79.21 | 154.60 |
| SCINet | 20.91 | 32.71 | 24.46 | 41.27 | 0.70 | 1.38 | 0.75 | 1.49 | 142.77 | 244.95 | 55.20 | 112.57 |
| STTN | 19.83 | 30.60 | 30.86 | 49.03 | 0.64 | 1.33 | OOM | OOM | 124.95 | 224.01 | OOM | OOM |
| STAEformer | 18.73 | 29.29 | 29.85 | 46.62 | 0.53 | 1.11 | OOM | OOM | 62.96 | 106.77 | OOM | OOM |
| **FlowNet** | **18.48**$^*$ | **29.03** | **22.79**$^*$ | **38.21**$^*$ | **0.43**$^*$ | **0.93**$^*$ | **0.50** | **1.08** | **39.03**$^*$ | **76.04**$^*$ | **31.39** | **59.06** |

## 4.3 Ablation Study (RQ2)

- **Effects of Retained Flow**. We remove the flow tokens $\Phi_o$ retained by each node, and each node's tokens are obtained only by the other nodes sending to themselves in this experiment. According to the results presented in Table 2, eliminating the retained flow has the least impact on the short-term forecasting task compared to eliminating the other items. However, when it comes to the long-term forecasting task, eliminating this item has a greater impact on FlowNet. Given that retained flow complements the flow tokens of the nodes, this suggests that the spatio-temporal system as a whole remains closed and that the total amount of information essentially remains unchanged in the short term. On longer time scales, however, the spatio-temporal system may experience additional inflows and outflows of information.

- **Effects of Allocation Flow**. We remove the flow tokens $\Phi_a$ that nodes send to each other, and the tokens for each node are only generated by the retained flow, which means that no more information is exchanged between nodes in this experiment. Eliminating this item has the greatest impact on the model, as can be observed in Table 2. This suggests that, in complex spatio-temporal systems, the exchange of information between nodes cannot be ignored, and that it is undesirable to use node independence alone to predict the system's evolution.

- **Effects of Conservation Laws**. We break the conservation law of flow exchange in this experiment, where each node no longer subtracts the flow tokens it sends after sending them. As shown in Table 2, breaking conservation laws has less impact on the model in long-term forecasting than eliminating retained flow. However, it has more impact in short-term forecasting. Combining this with previous analyses leads to an interesting conclusion: For short-term prediction, the spatio-temporal system is relatively closed. Therefore, maintaining information conservation

throughout the system, as well as in the exchange of information between nodes, is more important for predicting the system's evolution. For long-term prediction, however, information flowing into and out of the system cannot be ignored. It is more important to consider the fluctuation of the total amount of information in the system as a whole for accurate prediction.

Table 2: Ablation experiments on the PEMS04F dataset. $\Delta$ represents how much worse the results have become compared to the original model (a smaller value means better performance). The **bold**/underlined font means the worst/second-worst result.

| Task | Short-term | | | | | | Long-term | | | | | |
|---|---|---|---|---|---|---|---|---|---|---|---|---|
| w/o | Retained | | Allocation | | Conservation | | Retained | | Allocation | | Conservation | |
| Metrics | MAE | RMSE | MAE | RMSE | MAE | RMSE | MAE | RMSE | MAE | RMSE | MAE | RMSE |
| FlowNet | 18.56 | 29.16 | 24.02 | 36.94 | 18.72 | 29.36 | 24.11 | 39.95 | 27.69 | 45.47 | 23.62 | 39.65 |
| $\Delta$ (%) | 0.08 | 0.13 | **5.54** | **7.91** | 0.24 | 0.33 | 1.32 | 1.74 | **4.90** | **7.26** | 0.83 | 1.44 |

## 4.4 Efficiency Analysis (RQ3)

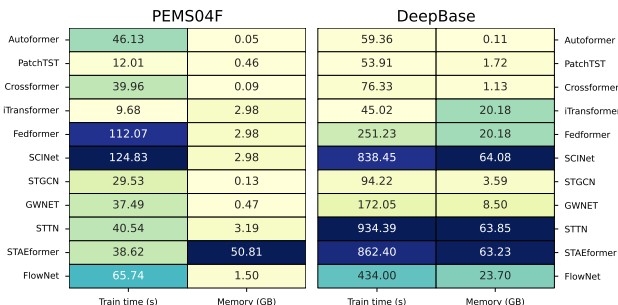

Figure 3: Efficiency analysis.

In this experiment, we analyze the training efficiency and GPU memory consumption of FlowNet versus baseline models on PEMS04F and DeepBase datasets for short-term forecasting tasks. As shown in Figure 3, FlowNet achieves intermediate computational efficiency between Transformer-based and STGNN-based models, balancing both training time and memory requirements. This efficiency profile stems from FlowNet's hybrid architecture that strategically integrates flow propagation mechanism with adaptive spatial masking, effectively balancing the computational intensity of full-sequence transformers against the memory overhead inherent in graph neural operations. Meanwhile, the advanced cascading method of hyper-connection does not significantly increase memory overhead. We skillfully achieve a balance between training efficiency and accuracy.

## 4.5 Distribution Analysis (RQ4)

In this experiment, we count the distribution of the degree and the perceptual radius of the nodes learned by ASM on the PEMS04F dataset. If node A is within the perceptual radius of node B, then we consider that there exists a directed edge from node B pointing to node A. As shown in Figure 6, for the long-term forecasting task, both the degree and perceptual radius distributions of the nodes are more concentrated compared to the

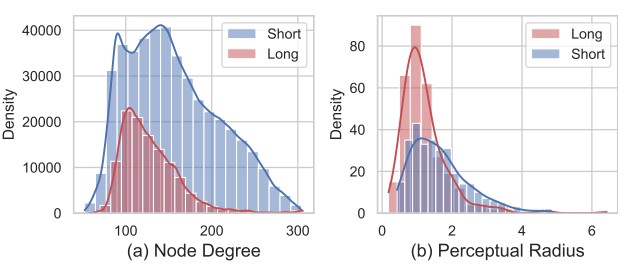

Figure 4: Distribution of node degree and perceptual radius.

short-term forecasting task. It is worth noting that the distributions of both the degree and radius of the nodes within the short-term prediction show a multi-peaked distribution. Based on this observation, we can reveal the reason for the sub-optimal performance of graph-based and attention-based models: graph-based models are unable to dynamically take into account dependencies between nodes according to diverse spatio-temporal systems and different tasks, while attention-based models compute extensive pseudo-dependencies, but dependencies exist among only some but not all nodes. Additionally, based on the previous analyses, both approaches mistake superficial proximity for intrinsic dependency, limiting the upper bound of the model's representation.

### 4.6 Visualization of Allocation Matrix (RQ5)

In Figure 5, we visualized the allocation matrix $\Lambda$ learned by FAM and compared it with the attention maps of transformer-based models such as iTransformer and STTN in the PEMS04F dataset. For FlowNet, we can observe that the allocation matrix of nodes is more centralized and clearer in the long-term prediction task than in the short-term prediction task. Based on the learning from FlowNet, some nodes receive the vast majority of the flow converged from the source nodes when they act as target nodes. In contrast, the attention maps of iTransformer and STTN have larger values for the attention coefficient, and the similarities captured between

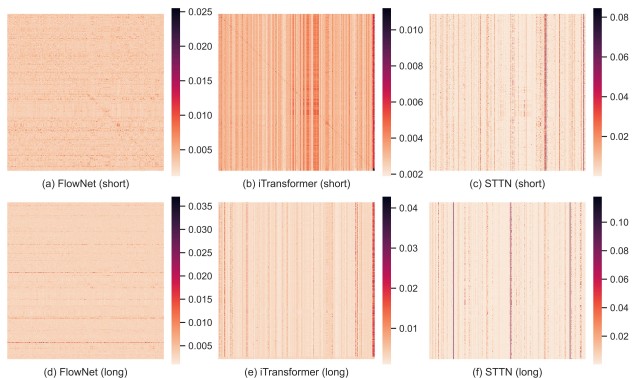

Figure 5: Heat map of the allocation matrix and the attention map learned by iTransformer and STTN. Each row and column represents a different node in the heat map, .

nodes are much denser. We speculate that this is due to the adaptive radius of the ASM filtering out many spurious dependencies. According to the statistics in Table 1, FlowNet outperforms both transformer-based models on all datasets, especially on DeepBase and SINPA, which are two large spatio-temporal systems. This suggests that filtering out spurious correlations is important to improve the performance of the model.

## 5 Related Works

Classical statistical methods [47, 48] model temporal dependencies through linear or probabilistic formulations, but fail to capture nonlinear spatio-temporal couplings. Recent innovations attempt to bridge these gaps through adaptive graph learning and hybrid architectures. Modern deep learning advances [22, 49] decompose temporal patterns via specialized mechanisms. Methods such as STS-GCN [26] and DCRNN [50] introduce dynamic graph structure adjustments while retaining heuristic similarity metrics. Transformer-based architectures like Crossformer [43] and iTransformer [44] advance cross-variable interaction modeling through dimension-aware attention or inverted embedding strategies. Concurrent work like STAEformer [19] incorporates graph information bottlenecks to improve interpretability. PDFormer [51] introduces a novel propagation delay-aware dynamic long-range dependency modeling approach through gated self-attention mechanisms, achieving advanced performance across multiple traffic flow prediction benchmarks. UniST [52] introduces a unified spatio-temporal learning framework that integrates adaptive multi-scale attention and task-agnostic representation learning, achieving superior accuracy and generalization across diverse spatio-temporal forecasting tasks. However, these approaches still neglect the directional flow dynamics inherent in transportation networks [53, 54] and hydrological systems [55, 56, 57]. Parallel progress in physics-inspired paradigms [58] prioritizes transfer mechanisms through spatio-temporal dynamic processes. However, such methods lack modular frameworks for adaptive propagation that could systematically model context-dependent information flows.

## 6 Conclusion and Future Work

In this work, we introduce FlowNet, a novel paradigm for modeling dynamic spatio-temporal systems that rethinks interactions in complex systems through the lens of directional flow dynamics. By shifting from static or similarity-driven representations to adaptive spatio-temporal flows, FlowNet overcomes critical limitations of existing methods. Experiments demonstrate FlowNet's superior accuracy and physical plausibility. Looking ahead, extending FlowNet to incorporate domain-specific conservation laws and applying it to flow-driven systems like supply chains or social networks presents promising directions. By harmonizing data-driven learning with principles of physical realism, this work advances the development of spatio-temporal models that better align with the dynamic, directional nature of real-world systems.

## Acknowledgments and Disclosure of Funding

This work is mainly supported by the Guangdong Basic and Applied Basic Research Foundation (No. 2025A1515011994). This work is also supported by the National Natural Science Foundation of China (No. 62402414), Tencent (CCF-Tencent Open Fund, Tencent Rhino-Bird Focused Research Program), Didi (CCF-DiDi GAIA Collaborative Research Funds), Guangzhou Municipal Science and Technology Project (No. 2023A03J0011), Huawei Industrial Funds, and the Guangzhou Industrial Information and Intelligent Key Laboratory Project (No. 2024A03J0628).

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

# A    More Details of Model Implementation

## A.1    Initialization

For the linear layers in M-MLP, we employed Kaiming initialization to set their initial parameters. All other linear layers in the model were initialized using Xavier initialization. GeLU is employed as the activation function in M-MLP. Specifically, for linear layers in the ASM module, we initialized the weight parameters $w$ to zero and set the bias terms $b$ to the average distance matrix for each corresponding dataset. Through this initialization method, each node can have a sufficiently large and identical receptive field at the beginning of training, and can optimize its own perception radius through gradient descent and backpropagation.

# B    More Details of Experiments

## B.1    Datasets

- **PEMS04F.** The PEMS04 [36] dataset, constructed from the Caltrans Performance Measurement System (PeMS), captures traffic flow dynamics across 307 sensor nodes in California over a two-month period (January 1 to February 28, 2018) with 5-minute granularity (16,992 timesteps). The original dataset includes three key traffic metrics (traffic flow, lane occupancy, and average speed), and we focus on the traffic flow (denoted as PEMS04F) attribute as the prediction target.

- **DeepBase.** The DeepBase [39] dataset is a hydrological dataset providing daily baseflow estimates for 1,661 basins across the contiguous United States (CONUS) from 1981 to 2022. This dataset captures the slow-varying groundwater contributions to streamflow at a daily temporal resolution.

- **SINPA.** The SINPA dataset [40] captures large-scale parking availability dynamics across Singapore, covering 1,687 parking lots over a one-year period (July 2020 – June 2021). It provides high-frequency spatio-temporal observations recorded at 15-minute intervals, where each node represents the available parking spaces at a specific lot. While the original dataset integrates diverse urban features, we focuses on modeling parking vacancy counts as the primary prediction target.

Table 3: Description of the datasets.

| Dataset | Category | Data Type | #Nodes | #Time points | Resolution | Date Range |
|---|---|---|---|---|---|---|
| PEMS04F | Traffic | Traffic flow | 307 | 16992 | 5 min | Jan.1, 2018 - Feb.28, 2018 |
| DeepBase | Hydrology | Base flow | 1661 | 14975 | 1 day | Jan.1, 1981 - Dec.31, 2022 |
| SINPA | Urban Mobility | Parking slot | 1687 | 35040 | 15 min | Jul.1, 2020 - Jun.30, 2021 |

## B.2    Preprocess

For the graph-based baseline, we construct a weighted adjacency matrix by applying a thresholded Gaussian kernel to pairwise Euclidean distances between nodes:

$$
W_{ij} = \begin{cases} \exp\left(-\frac{\mathrm{dist}(v_i,v_j)^2}{\sigma^2}\right), & \text{if dist}\,(v_i,v_j) \leq \kappa \\ 0, & \text{otherwise} \end{cases}
$$

where edge weight $W_{ij} \in [0,1]$ encodes proximity between stations $v_i$ and $v_j$ , $\sigma$ controls the kernel width, and $\kappa$ sparsifies connections beyond local neighborhoods.

We standardize the spatiotemporal dataset $\mathbf{X} \in \mathbb{R}^{N \times T}$ ($N$ nodes, $T$ timesteps) via z-score normalization:

$$
\tilde{x}_{i,t} = \frac{x_{i,t} - \mu_i}{\sigma_i} \quad \forall i \in \{1,\ldots,N\},
$$

where $\mu \in \mathbb{R}^N$ and $\sigma \in \mathbb{R}^N$ denote per-node training means and standard deviations. During evaluation, predictions on validation/test sets are inversely transformed $\hat{x}_{i,t} = \tilde{x}_{i,t} \cdot \sigma_i + \mu_i$ before computing evaluation metrics.

**B.3 Training & Validation**

We configure prediction horizons based on temporal granularity and dominant frequencies across datasets. For PEMS04F and SINPA, short-term forecasting uses 12-step input/output sequences (1h/3h), while long-term forecasting employs 288-step windows (1d/3d). DeepBase adopts 32-step (monthly, ensuring divisibility for patching) and 360-step (yearly) horizons, respectively. The datasets are partitioned as follows: PEMS04F (70% train, 10% validation, 20% test), DeepBase (pre-2021 train, 2021-2022 validation/test), and SINPA (pre-May-2021 train, May-June 2021 validation/test). We implement sliding window sampling with stride 1 during training, aligning window size with output horizon during inference.

Following standard evaluation protocols in machine learning, we quantify the predictive performance of regression models through two widely adopted metrics: the Mean Absolute Error (MAE) and Root Mean Squared Error (RMSE). Formally, let $Y_i \in \mathbb{R}$ represent the ground-truth value of the $i$-th data instance and $\hat{Y}_i \in \mathbb{R}$ denote its corresponding predicted value. These error metrics are computed as

$$\text{MAE} = \frac{1}{n} \sum_{i=1}^{n} |Y_i - \hat{Y}_i| \tag{11}$$

$$\text{RMSE} = \sqrt{\frac{1}{n} \sum_{i=1}^{n} (Y_i - \hat{Y}_i)^2} \tag{12}$$

where $n$ denotes the total number of test samples. During training, we employ MAE as the loss function for FlowNet and all baselines. Our Early Stopping mode protocol monitors the MAE metric at the validation set and terminates training when no improvement is observed for 10 consecutive epochs relative to the best recorded value, indicating potential overfitting. We retain the model parameters, achieving the lowest validation MAE for final evaluation on the test set.

## C  Details of Baselines

- **Autoformer** [41]: Autoformer integrates decomposition architecture into Transformers, replacing self-attention with an auto-correlation mechanism to capture periodic dependencies. It progressively decomposes trend and seasonal components, achieving efficient long-term forecasting with linear complexity.

- **PatchTST** [42]: By segmenting time series into independent patches and adopting channel-independent modeling, PatchTST enhances local semantic extraction and reduces computational costs. It outperforms traditional Transformers in long-term forecasting tasks by leveraging vision transformer-inspired patch processing and self-supervised learning.

- **Crossformer** [43]: Crossformer employs a two-stage attention mechanism to model cross-dimension dependencies in multivariate time series. Its hierarchical encoder-decoder structure and dimension-segment-wise embedding efficiently capture interactions between time steps and variables.

- **iTransformer** [44]: iTransformer inverts the standard architecture by treating time points as tokens and applying attention across variables.

- **FEDformer** [45]: Combining frequency-domain transformations with seasonal-trend decomposition, FEDformer reduces computational overhead through random frequency component selection. Its linear complexity and frequency-enhanced blocks make it effective for long-term forecasting in energy and weather datasets.

- **SCINet** [46]: SCINet uses a binary tree structure with interactive convolution blocks to hierarchically decompose time series. This architecture captures multi-resolution temporal dependencies and mitigates information loss, outperforming RNN and Transformer models in efficiency and accuracy.

- **STGCN** [25]: STGCN integrates graph convolutional networks (GCNs) and gated temporal convolutions to model traffic networks. Its fully convolutional design processes large-scale spatiotemporal data efficiently.

- **GWNET** [35]: This model introduces an adaptive adjacency matrix to learn hidden spatial dependencies and dilated convolutions for long-range temporal patterns. It addresses incomplete graph structures in traffic forecasting and achieves linear complexity with stable multi-step predictions.

- **STTN** [16]: Spatial-Temporal Transformer Network replaces GCNs with dynamic spatial attention to capture time-varying node relationships. Its non-autoregressive multi-step prediction framework avoids error accumulation, significantly improving long-horizon forecasting.

- **STAEformer** [19]: By incorporating spatiotemporal adaptive embeddings into vanilla Transformers, STAEformer dynamically adjusts to traffic patterns without complex architectural modifications by preserving intrinsic chronological information and spatial heterogeneity through lightweight adaptive components.

All baseline code implementations are based on the time-series-library [59] and LargeST [60]. Specifically, to ensure the fairness of the comparison and guarantee that most models can run on our existing computing resources, we set the hidden dimension of all models to 64. For encoder-only models, we stacked 2 encoder layers. For encoder-decoder architecture models, we stacked 1 encoder layer and 1 decoder layer.

## D    Further Analysis

We analyze task-specific differences in node degree and perceptual radius distributions for short- and long-term predictions on the SINPA dataset. Short-term predictions exhibit a sharp node degree distribution peaking, indicating reliance on highly connected nodes to capture dense local interactions. In contrast, long-term predictions show a flatter distribution peaking below 500, suggesting sparser node sampling to prioritize broader contextual patterns over fine-grained dynamics. Short-term predictions concentrate around moderate radii, balancing localized state changes and mid-range dependencies. Long-term predictions, however, favor smaller radii with greater dispersion, reflecting adaptive trade-offs: smaller radii suppress noise from distant nodes, while occasional long-range connections capture systemic shifts, such as holiday-driven destination changes. These divergences reveal that the model dynamically tailors its spatial aggregation strategy to temporal scope: short-term tasks emphasize resolution-rich local dynamics, whereas long-term tasks hierarchically integrate multi-scale dependencies at lower resolution. The distinct distributions further validate the necessity of task-aware spatial perception mechanisms in spatio-temporal forecasting.

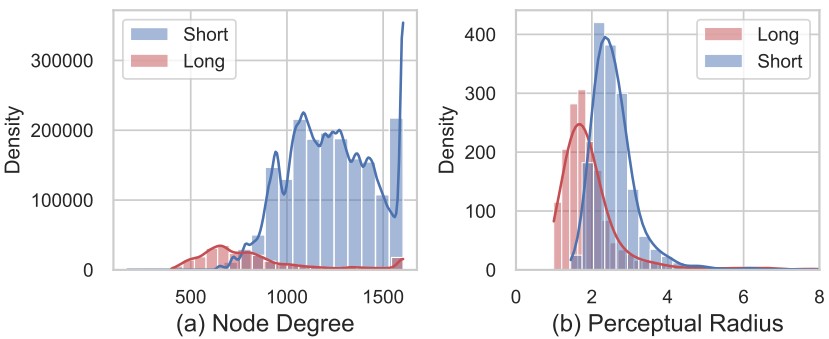

Figure 6: Node property distributions for short-term (blue) and long-term (red) forecasting.

## E    More Discussion

**Limitations.** While FlowNet achieves state-of-the-art prediction accuracy, its computational efficiency is constrained by pairwise operations. Specifically, the Flow Allocation operation incurs an $O(N^2)$ complexity due to node-wise flow redistribution, and the Flow Evolution Module (FEM) scales quadratically with the prediction horizon ($O(T^2)$). For systems with large node counts or long-term forecasting tasks, FlowNet remains less efficient than STGNN-based methods, though it outperforms Transformer-based models in both speed and memory usage. We argue that the accuracy

gains justify this trade-off in many real-world applications. Future work will explore sparse flow tokenization and hierarchical grouping to mitigate these bottlenecks.

**Societal Impacts.** FlowNet's ability to model flow-driven dynamics can benefit urban planning, environmental protection, and disaster response. The framework enhances interpretability by explicitly quantifying flow exchanges, enabling policymakers to design targeted regulations. Furthermore, its conservation-law alignment promotes physically plausible predictions, reducing risks of harmful decisions based on spurious correlations.

