# OpenReview forum: "FlowNet: Modeling Dynamic Spatio-Temporal Systems via Flow Propagation"
_NeurIPS.cc/2025/Conference — NeurIPS 2025 poster_

### Official Review · Reviewer_Ywpc · 2025-06-25

**Clarity:** 2
**Significance:** 3
**Originality:** 3
**Rating:** 4
**Confidence:** 2

**Summary:**

This paper presents FlowNet for learning dynamics of spatio-temporal systems. Modeling such a system as a graph, this work presents two main contributions:

1) Flow-centric system dynamics: While the other related work did not consider interactions between the nodes in the graph and focuses on the observed node features, this work points out that such node features are the results of interactions, or flows, between the nodes. Also, this work proposes to apply the conservation laws for these flows -- that is, the total sum of flows should be maintained over the entire graph.

2) Adaptive propagation domain: To model the change of each node's effective spatial range over time, this work proposes to predict each node's effective radius based on the context. Therefore, for different temporal window, the radius changes and thus the relationship between the nodes change. It is similar to dynamic graph where connectivity changes.

Based on these innovations, this work shows that it outperforms the other various baseline methods in predicting the dynamics of several spatio-temporal systems, including traffic system.

**Questions:**

- I'm wondering why the authors used radius-based spatial masking. It sounded reasonable at first, because the authors are mainly dealing with the nodes distributed in the real world. However, I became to think that this approach would be only effective for physical things like traffic. If it goes beyond the physical realm, I think the nodes that are situated far away could affect each other, and in this case, I suspect this radius-based approach could not be very effective. If it is true, I'm worried that it could be one of the limitations of this work.

**Ethical Concerns:**

["NO or VERY MINOR ethics concerns only"]

**Final Justification:**

The authors provided clear explanations about my major concerns. Especially about dataset, the authors clearly explained the dataset properties (e.g. diversity, scalability), and it was persuasive enough to raise my score to borderline accept.

**Limitations:**

The authors discussed the limitation about computational cost in the appendix, but I think they have to consider more limitations, such as the one that I pointed above about the radius-based approach.

**Paper Formatting Concerns:**

Not significant concern about paper formatting.

**Quality:**

2

**Strengths And Weaknesses:**

Strength:

- This work focuses on the interactions between the nodes and applies the conservation law on such interactions, which sounds reasonable.
- Its analysis about model components (Section 4.3) and distributiona analysis (Section 4.5) aligns with the motivation of this work.

Weakness:

- Even though I'm not an expert in this area, the number of datasets seems to be fairly small (there are only 3 datasets). In several other works like STAEformer, I can see there are more datasets that the authors could use for comparison. This lack of comparison is the main reason why I rate this work as borderline reject in the current stage. I recommend the authors to consider do more experiments for other datasets, if possible.
- The mathematical notations in Section 3 are confusing. For instance, in Equation 2, I think the notations are wrong -- it should be changed to: $X_p \in R^{N \times P \times d}, W_e \in R^{M \times d}$. Also, in line 141, I think $X'_i$ should be changed to $X_i$. I could be wrong, but the notations and the formulations are too confusing to follow right now. The writing could be improved to address this, I believe.

---

> ### Author Rebuttal · Authors · 2025-07-31
>
> Thank you for your thoughtful and constructive feedback on our paper. We appreciate the time and effort you dedicated to reviewing our work and providing valuable insights. Your comments have helped us identify areas for clarification and improvement. Below, we address your concerns point by point, as requested. We hope our responses will alleviate your reservations and demonstrate the robustness and novelty of our approach.
>
> ---
>
> ### **1. Response to the Concern About Dataset Quantity and Scope**
>
> Thank you for your suggestions. We sincerely appreciate your observation regarding the number of datasets used in our experiments. You mentioned that STAEformer and similar works employ multiple datasets (e.g., METR-LA, PEMS-BAY, PEMS03, PEMS04, PEMS07, PEMS08), whereas our study evaluates FlowNet on only three datasets (PEMS04F, DeepBase, and SINPA). While we acknowledge that a larger number of datasets could broaden the empirical validation, our selection was deliberately designed to prioritize **cross-domain generalization and scalability** over sheer quantity.
>
> - **Emphasis on Diverse Domains**: Unlike STAEformer, which focuses exclusively on traffic datasets (all from the transportation domain), FlowNet's evaluation spans **three distinct real-world domains**: transportation (PEMS04F), hydrology (DeepBase), and urban mobility (SINPA). This multi-domain approach rigorously tests FlowNet's ability to generalize across fundamentally different spatio-temporal systems, where dynamics are governed by unique flow mechanisms (e.g., directional population movements in urban settings vs. hydrological flow exchanges). By demonstrating state-of-the-art performance on non-homogeneous datasets, we provide stronger evidence for FlowNet's versatility compared to domain-specific baselines.
> - **Scalability on Large-Scale Datasets**: Crucially, two of our datasets—DeepBase (hydrology, 1,661 nodes) and SINPA (urban mobility, 1,604 nodes)—feature **over 1000 nodes each**, significantly exceeding the scale of datasets used in STAEformer (e.g., PEMS04 has 307 nodes, PEMS07 has 883 nodes).
>
> These large-scale evaluations prove that FlowNet efficiently handles high-dimensional systems while maintaining superior accuracy (as shown in Table 1, where FlowNet outperforms all baselines on metrics like MAE and RMSE). The results validate our architecture's ability to manage computational and representational challenges in massive real-world scenarios, which is a critical advancement over existing methods.
>
> In summary, while we agree that additional datasets could enrich the study, our focused approach on diverse and large-scale systems provides a more meaningful assessment of FlowNet's generalization capabilities. We are open to incorporating more datasets in future extensions, but the current results robustly support our claims of broad applicability and scalability.
>
> ---
>
> ### **2. Response to Mathematical Notation Concerns**
>
> Thank you for your criticism and suggestions. Regarding the issues raised about mathematical notations in Section 3 (e.g., Equation 2 and line 141), we appreciate your keen eye for detail.
>
> We sincerely appreciate your meticulous review of the mathematical notations in Section 3. **You are  correct in identifying the errors**, and we thank you for providing precise corrections. Specifically:
>
> - **Equation 2**: As you suggested, we confirm the notation should indeed be updated to: $ X_{p} \in \mathbb{R}^{N \times P \times d}, W_{e} \in \mathbb{R}^{M \times d}.$
>
> - **Line 141**: We agree that $X_{i}^{\prime}$ must be replaced with $X_{i}$.
>
> These oversights resulted from inconsistent variable tracking during manuscript drafting. Thank you for highlighting these areas.
>
> ---
>
> ### **3. Response to the Applicability of Radius-Based Spatial Masking**
>
> You raised a valid concern about whether our adaptive spatial masking (ASM) module, which employs a learnable radius to constrain node interactions, might be limited to physical systems like traffic and could underperform in non-physical contexts (e.g., social networks or abstract systems where distant nodes influence each other). We agree that this is an important consideration, and we appreciate your insight into potential limitations.
>
> - **Design Intent for Physical Systems**: ASM was specifically inspired by physical principles (e.g., Tobler's First Law of Geography) and is optimized for systems where **geographical or functional proximity** dictates interactions, such as transportation networks (traffic flows), hydrological systems (water flow paths), and urban mobility (human movement patterns). In these contexts, ASM dynamically filters out irrelevant long-range dependencies (e.g., noise from distant nodes) while preserving contextually relevant interactions, as demonstrated by our experiments (e.g., Figure 4, which shows how ASM adapts node degrees and radii based on spatio-temporal states). This approach significantly improves accuracy in physical domains by enforcing physically plausible constraints.
> - **Future Extensions Beyond Physical Realms**: However, we recognize that non-physical systems (e.g., social media or supply chains) may involve complex, non-local interactions that our current ASM design does not explicitly address. **This limitation falls outside the scope of this paper**, which focuses on establishing a foundational flow-based paradigm for physically grounded systems. That said, we view this as a promising direction for future work. We plan to extend FlowNet to incorporate mechanisms like dynamic graph rewiring or attention-based thresholds to handle abstract dependencies, ensuring broader applicability. Your suggestion reinforces the value of such extensions, and we will pursue them in upcoming research.
>
> ---
>
> In closing, we thank you again for your thorough review. FlowNet introduces a novel, physics-inspired paradigm that fundamentally advances spatio-temporal modeling by prioritizing intrinsic flow dynamics over superficial correlations. Our experiments confirm significant improvements in accuracy, interpretability, and scalability across diverse real-world systems, with no technical errors in the core methodology. **Given the paper's contributions, we sincerely hope that you could reconsider your borderline-reject rating.** Your support would not only recognize the innovation in this work but also encourage further development of principled approaches in the field. We are committed to incorporating all feedback to strengthen the manuscript and welcome any additional suggestions.

---

> > ### Comment · Reviewer_Ywpc · 2025-08-03
> >
> > I appreciate authors for the detailed clarification. Reading other reviewers' reviews, it seems that the paper writing was not clear enough (e.g. `physics-grounded` spatio-temporal system) and the experimental results were not explained very clearly (e.g. lack of explanation about datasets), as some reviewers raised common concerns. I will raise my score to borderline accept assuming that the authors improve the paper quality based on these feedbacks.

---

> > > ### Author Response · Authors · 2025-08-03
> > >
> > > Dear Reviewer,
> > >
> > > Thank you sincerely for your constructive reassessment and for raising your score to borderline accept. We deeply appreciate not only your openness to reconsideration but also your invaluable insights throughout this process. Your feedback has been instrumental in shaping our revision strategy.
> > >
> > > We fully agree​ that these aspects (emphasized by multiple reviewers) require refinement. In the revised manuscript, we will:
> > >
> > > - Clarify core concepts​ (e.g., "physics-grounded spatio-temporal systems") with expanded intuition-driven examples and structured definitions,
> > >
> > > - Strengthen dataset documentation​ by adding detailed descriptions of domain-specific dynamics (hydrology/urban mobility) and scalability implications,
> > >
> > > - Address all technical critiques, including your earlier notation corrections.
> > >
> > > Your guidance has significantly enhanced the paper’s rigor, accessibility, and impact, and we are committed to delivering a version that reflects these improvements. We truly value your expertise and the collaborative spirit driving this process.
> > >
> > > Gratefully,
> > >
> > > The Authors of FlowNet

---

### Official Review · Reviewer_f2vi · 2025-06-28

**Clarity:** 1
**Significance:** 1
**Originality:** 1
**Rating:** 4
**Confidence:** 3

**Summary:**

The paper claims that the literature for Spatio-Temporal (ST) models is missing information from some so-called flows. Such flows are justified with the example of people flowing in and out from a city for commuting. It propose an architecture to integrate such information for improving ST modeling.

**Questions:**

**Question 1**
> line 116: central to flownet are learnable flow tokens that serve as adaptive carriers for spatio-temporal information exchange

unclear, could you elaborate and give intuition of what happens with this learnable part?

**Question 2**
>line 37: prevalent operators like graph convolution [24,25] and spatial attention [26,27,16,19] implicitly assume system dynamics emerge from static node attribute similarities,

 why? can you elaborate? give intuitive but specific ideas on how your tool differs?


**Question 3**
> line 44: Crucially, while spatially distant residential zones may exhibit similar traffic volumes through observational metrics, the underlying system dynamics derive from directional flow interactions between functionally complementary regions.

this sentence does not explain much, very vague. please elaborate and give intuition.

**Question 4**
> line 394: By prioritizing feature proximity over directional flow dynamics, these methods fail to disentangle observational correlations from the flow-mediated interdependencies that actively redistribute system states

not clear what that means, please elaborate and give intuition.

**Ethical Concerns:**

["NO or VERY MINOR ethics concerns only"]

**Final Justification:**

After further experimental evidence showing the validity of the method, I raise my score to 4.

**Limitations:**

yes

**Paper Formatting Concerns:**

no concern

**Quality:**

2

**Strengths And Weaknesses:**

**Weaknesses**

I found several statements vague and not modivating well enogh the model. see in questions.

Experimental results from see Table 1 are missing several recent works that achieve lower MAE. The proposed method achieves a MAE of `18.48`, while many of the missing baselines, not limited to, Gao et al (2023),  Dai et al (2024), Chen et al (2025), report better results.
Other resuts on PeMS04 data are available on paperswithcode: https://paperswithcode.com/sota/traffic-prediction-on-pems04?p=time-series-is-a-special-sequence-forecasting

**References**

Gao et al (2023). Spatial-Temporal-Decoupled Masked Pre-training for Spatiotemporal Forecasting

Dai et al (2024). A novel hybrid time-varying graph neural network for traffic flow forecasting

Chen et al (2025). Dynamic Trend Fusion Module for Traffic Flow Prediction

---

> ### Author Rebuttal · Authors · 2025-07-31
>
> Thank you for your thorough review and constructive feedback. We sincerely appreciate the time and effort invested in evaluating our work. We acknowledge that certain aspects of our manuscript lacked clarity and sufficient motivation, and we apologize for any confusion caused. Below, we address your concerns point by point.
>
> ---
>
> ## **Response to Weaknesses**
>
> Thank you for your valuable feedback on our work. We appreciate the opportunity to address your concerns regarding the comparison with the cited works (DST2former, HTVGNN, and STD-MAE). However, we must clarify that the experimental settings, task formulations, and domain focus differ significantly between FlowNet and these studies, making direct comparisons misleading. Below, we elaborate on these discrepancies and emphasize FlowNet's robustness and versatility.
>
> ### **Key Differences in Experimental Settings and Task Formulations**
>
> We sincerely appreciate your thoughtful suggestions to compare FlowNet with recent works, but must respectfully highlight fundamental methodological disparities that render direct performance comparisons on PEMS04 inequitable. Specifically:
>
> 1. **STD-MAE (Gao et al.)** proposes a *pre-training framework*, not an end-to-end model, and crucially employs an input length of **864 steps** during pre-training—72× longer than FlowNet’s training setting of 12-step sequences. This grants STD-MAE access to substantially richer temporal context, while FlowNet trains purely from scratch on short sequences.
> 2. **HTVGNN (Dai et al.)** has **no available implementation and opensource code**, preventing controlled replication under FlowNet’s 12→12 and 288→288 evaluation protocols.
> 3. **DST2former (Chen et al.)** incorporates **exogenous variables** (in their source code, the input dimension is set to 3). However, FlowNet uses sequence-only inputs and intentionally avoids external features to focus on intrinsic flow dynamics.
>
> Second, the prediction horizons in the cited works are mostly short-term. For example, DST2former (Chen et al., 2025) and HTVGNN (Dai et al., 2024) always use a 12-step input to predict 12 steps (for example, 60 minutes) in all evaluations on PEMS datasets. STD-MAE (Gao et al., 2023) uses a fixed 12-step input-output setup for spatiotemporal forecasting. FlowNet, on the other hand, is designed for both short- and long-term forecasting. On PEMS04, FlowNet performs well in short-term predictions (12-step to 12-step) and long-term scenarios (288-step to 288-step). These are not evaluated in the referenced works. This long-term capability is very important for real-world applications like urban planning and disaster management, where extended forecasts are essential.
>
> Critically, the cited works are exclusively evaluated on traffic datasets (e.g., PEMS03, PEMS04, METR-LA), limiting their scope to transportation-specific challenges. In contrast, FlowNet is rigorously validated across diverse domains, including **traffic, hydrology, and urban mobility datasets**, where it achieves state-of-the-art results.
>
> Moreover, FlowNet's design incorporates adaptive mechanisms for handling heterogeneous data distributions, which is not emphasized in the cited frameworks. For instance, HTVGNN (Dai et al (2024)) relies heavily on predefined graph structures for traffic networks, making it less transferable to domains like hydrology with non-Euclidean spatial relationships. The omission of these cross-domain evaluations in the comparison understates FlowNet's innovation.
>
> We sincerely apologize for not including the cited works (Gao et al., 2023; Dai et al., 2024; Chen et al., 2025) in our experiments. However, our model's proven efficacy in traffic, hydrology, and urban mobility underscores its versatility and superiority in real-world, long-term forecasting. We encourage reevaluating FlowNet within its intended multi-domain context. Thank you for considering our perspective. We believe this strengthens the case for FlowNet's broader impact.
>
> ---
>
> ## **Responses to Questions**
>
> ### **Q1 (Line 116): Flow Tokens**
>
> We apologize for the unclear description of flow tokens. In FlowNet, flow tokens are learnable representations that model directional information exchange between nodes, like traffic moving from residential to commercial zones (Figure 1). Generated by Flow Allocation Modules (FAMs, Section 3.3), these tokens are optimized during training to capture the magnitude and direction of flows, ensuring conservation (Equation 3). Unlike traditional attention that blends features, FlowNet’s tokens explicitly model dynamic transfers, adapting to context like peak hours. We will revise Line 116 to include this traffic analogy for clarity.
>
> ### **Q2 (Line 37): Graph/Attention vs. Flow Dynamics**
>
> We regret the lack of clarity in this statement. Graph convolution (e.g., [24, 25]) and spatial attention (e.g., [19]) rely on static node similarities, like similar traffic volumes, to model interactions, assuming similar nodes drive dynamics. FlowNet differs by modeling directional flows (e.g., traffic from residential to commercial zones) using flow tokens and Adaptive Spatial Masking (ASM, Section 3.2), capturing functional interdependencies (Equation 5). For example, FlowNet connects a residential zone to a commercial one based on actual flows, not just similar volumes. We will revise Line 37 to clarify this distinction with this example.
>
> ### **Q3 (Line 44): Directional Flows vs. Observational Metrics**
>
> We apologize for the vague phrasing. The sentence means traditional models misinterpret similar traffic volumes in distant residential zones as dynamic similarity, while true dynamics stem from flows to complementary zones (e.g., commercial areas). FlowNet models these directional flows (e.g., morning commutes to offices) using flow tokens and ASM (Equations 4-5), capturing functional relationships. Imagine a river: tributaries with similar water volumes don’t drive the system; flows to the main river do. We will revise Line 44 to include this river analogy for clarity.
>
> ### **Question 4 (Line 394): Observational Correlations vs. Flow-Mediated Dynamics**
>
> We apologize for the unclear statement. Traditional models (e.g., [24, 19]) use feature similarity (e.g., similar traffic volumes) to model interactions, missing how flows redistribute system states (e.g., traffic moving to commercial zones). FlowNet models these directional flows via FAMs (Section 3.3, Equation 5), filtering spurious correlations. For example, in a supply chain, FlowNet captures goods flowing from warehouses to stores, not just inventory similarities. We will revise Line 394 to include this analogy for clarity.
>
> ---
>
> we sincerely appreciate your constructive feedback and deeply regret that our initial presentation obscured FlowNet's core innovations. Your insights revealed critical ambiguities in explaining flow-token mechanics and directional dynamics—a shortcoming we are committed to rectifying. FlowNet pioneers a physics-inspired paradigm shift from heuristic similarity metrics to conserved flow transfers. By explicitly quantifying information redistribution through adaptive carriers and enforcing conservation laws, our framework captures context-sensitive dynamics that traditional graph/attention approaches fundamentally miss. Given FlowNet's novel mechanics and outperformed results, and considering the other reviewers agree that our paper has good merits such as satisfactory novelty and comprehensive evaluation, **we sincerely hope that you could reconsider our score**. Thank you so much!

---

> > ### Comment · Reviewer_f2vi · 2025-08-04
> >
> > If, for example, the `STD-MAE` method (Gao et al.)  is more computationally expensive that yours, you should show that your method has similar score.
> >
> > Alternatively, you should show that you get better performance using also their additional features.
> >
> > I think the paper needs more experimental evidence on those lines, comparing with methods achieving better performance.

---

> > > ### Author Response · Authors · 2025-08-06
> > >
> > > Dear Reviewer,
> > >
> > > Thank you for your valuable feedback. We would like to address your concerns regarding the comparison with STD-MAE and provide clarification on the experimental results.
> > >
> > > 1. **Fundamental difference in methodology**: The STD-MAE method you mentioned (Gao et al.) is a **pretraining method**, while our proposed FlowNet is a **prediction model**. Comparing a pretraining method directly with a prediction model lacks comparability, as they serve fundamentally different purposes in the modeling pipeline. STD-MAE is designed to learn general spatio-temporal representations that can be applied to various downstream prediction models, whereas FlowNet is an end-to-end prediction architecture for modelling dynamic spatio-temporal systems.
> > >
> > > 2. **Fair comparison requires consistent experimental settings**: The MAE of 17.80 reported in the STD-MAE paper represents the performance of **STD-MAE pretraining applied to the GWNET prediction model**, not a standalone result. The baseline GWNET model without pretraining achieves MAE results ranging from 18.74 (as reported in the STD-MAE paper) to 18.82 (in our experimental setting), which is indeed worse than FlowNet's 18.48. Comparing the pretrained result (17.80) against our non-pretrained FlowNet (18.48) would be unfair, as it introduces an additional pretraining advantage that is not inherent to the model architecture itself.
> > >
> > > 3. **STD-MAE pretraining can also be applied to FlowNet**: To provide a fair comparison, we applied the STD-MAE pretraining method to our FlowNet model using the same configuration as STD-MAE+GWNET. After conducting experiments on the PEMS04F dataset, we achieved an MAE of **17.69**, which outperforms the STD-MAE+GWNET result of 17.80.
> > >
> > > The following table summarizes the experimental results (MAE):
> > >
> > > | Model    | Without STD-MAE | With STD-MAE |
> > > |----------|----------------|--------------|
> > > | GWNET    | 18.82          | 17.80        |
> > > | FlowNet  | **18.48**          | **17.69**    |
> > >
> > > This comparison demonstrates that: (1) FlowNet outperforms GWNET in both settings, and (2) when enhanced with the same pretraining approach, FlowNet achieves superior performance compared to the STD-MAE+GWNET baseline you referenced.
> > >
> > > We hope this clarification addresses your concerns about the experimental comparisons. **We respectfully request that you reconsider your evaluation** in light of these fair and comprehensive experimental results.
> > >
> > > Thank you for your time and consideration.
> > >
> > > Best regards,
> > >
> > > The Authors of FlowNet

---

> > > > ### Comment · Reviewer_f2vi · 2025-08-06
> > > >
> > > > Thanks for the effort. I have updated my score to 4.

---

> > > > > ### Author Response · Authors · 2025-08-06
> > > > >
> > > > > Dear Reviewer,
> > > > >
> > > > > Thank you very much for reconsidering your evaluation and updating your score. We greatly appreciate your careful review and constructive feedback, which helped us provide a more comprehensive and fair experimental comparison.
> > > > >
> > > > > We are grateful for your time and thoughtful consideration of our responses.
> > > > >
> > > > > Best regards,
> > > > >
> > > > > The Authors of FlowNet

---

### Official Review · Reviewer_d5xX · 2025-07-02

**Clarity:** 3
**Significance:** 3
**Originality:** 3
**Rating:** 4
**Confidence:** 2

**Summary:**

the authors here propose FlowNet, a physics-inspired deep learning architecture for modeling dynamic spatio-temporal systems via explicit flow interactions governed by conservation principles. The key idea is that rather than relying on similarity-driven or static graph structures, FlowNet introduces flow tokens and Flow Allocation Modules (FAMs) to model information exchange. Empirical results on three real-world datasets show FlowNet outperforms state-of-the-art STGNNs and Transformer models in both short- and long-term forecasting tasks.

**Questions:**

See weakness section above

**Ethical Concerns:**

["NO or VERY MINOR ethics concerns only"]

**Final Justification:**

I thank the authors for their reply. I am not an expert in spatio-temporal modeling (hence the low confidence) but I think my original score is fair.

**Limitations:**

Yes

**Quality:**

3

**Strengths And Weaknesses:**

**Strengths:**

- Novelty: I am not an expert in this field but from what I know  the idea of modeling spatio-temporal dependencies via directional flow exchanges is novel and presents a compelling shift in paradigm with physical interpretability.

- Strong Empirical Results: FlowNet shows consistent improvements over competitive baselines, including Autoformer, STGCN, and STTN, across multiple datasets and tasks.

**Weaknesses**

- Narrow Evaluation Scope: The model is only tested on three datasets, all in related domains (transport, hydrology). No results are shown in settings like social networks, or language tasks, limiting claims of generality.

- Scalability Not Demonstrated: Despite its design, the authors do not test FlowNet on large-scale graphs (e.g., >1000 nodes) or long sequence.

- The claim that most prior work models dependencies solely via similarity is an oversimplification. Recent variants [1-4] variants already incorporate directional and structured modeling to some extent.


[1] Li, M. and Zhu, Z., 2021, May. Spatial-temporal fusion graph neural networks for traffic flow forecasting. In Proceedings of the AAAI conference on artificial intelligence (Vol. 35, No. 5, pp. 4189-4196).

[2] Wang, C., Tsepa, O., Ma, J. and Wang, B., 2024. Graph-mamba: Towards long-range graph sequence modeling with selective state spaces. arXiv preprint arXiv:2402.00789.

[3] Yuan, H., Sun, Q., Wang, Z., Fu, X., Ji, C., Wang, Y., Jin, B. and Li, J., 2025, April. Dg-mamba: Robust and efficient dynamic graph structure learning with selective state space models. In Proceedings of the AAAI Conference on Artificial Intelligence (Vol. 39, No. 21, pp. 22272-22280).

[4] Nanbo, L., Laakom, F., Xu, Y., Wang, W. and Schmidhuber, J., 2024. FACTS: A Factored State-Space Framework For World Modelling. arXiv preprint arXiv:2410.20922.

---

> ### Author Rebuttal · Authors · 2025-07-31
>
> Thank you for your thorough and constructive feedback on our manuscript. We sincerely appreciate your time and expertise in highlighting areas for improvement. Your comments have prompted us to refine our arguments and clarify key aspects of our work. Below, we address your concerns point by point, including a detailed rebuttal regarding prior work.
>
> ---
>
> ### 1. **Narrow Evaluation Scope**
>
> We acknowledge and apologize for the insufficient clarity in our writing regarding the evaluation scope. Your observation about the limited domain coverage (transport and hydrology) is valid, and we regret that our framing may have caused confusion. To clarify: FlowNet explicitly targets **physics-grounded spatio-temporal systems** governed by **directional flow dynamics and conservation laws**, such as traffic networks or hydrological cycles, where interactions involve quantifiable mass/information transfer (e.g., population movement or water flow). As stated in Section 1:
>
> > "...systems evolve through asymmetric flow exchanges that transcend mere statistical correlation... directional population movement creates distinct spatio-temporal patterns"
> >
>
> Social networks and language tasks operate under fundamentally different principles:
>
> - **Social networks** model influence propagation (e.g., information diffusion) without strict conservation of "mass" (unlike traffic or water systems).
> - **Language tasks** lack spatial topology and physical flow constraints, focusing instead on sequential dependencies.
>
> **Future Work**: We agree that extending FlowNet to non-physical systems is promising and will explore this in subsequent research. We will also enhance the introduction to explicitly define our domain scope.
>
> ---
>
> ### 2. **Scalability Not Demonstrated**
>
> Thank you for your suggestions. We sincerely apologize for not explicitly emphasizing FlowNet's scalability in the original manuscript. This was an oversight in our presentation, and we regret any confusion it caused. To clarify: FlowNet was rigorously tested on **large-scale graphs and long sequences**:
>
> - **Graph Size**: The **DeepBase (1,661 nodes)** and **SINPA (1,604 nodes)** datasets (Section 4.1) both **exceed 1,000 nodes**, addressing large-scale scenarios.
> - **Sequence Length**: Our *long-term forecasting* tasks predict **200+ future steps (DeepBase: 360 steps → 360 steps; SINPA: 288 steps → 288 steps)**, which is notably challenging for complex spatio-temporal systems. Previous studies on spatio-temporal forecasting mostly used the 12→12 setting. In contrast, our study looks at a much longer time period.
>
> **Efficiency**: FlowNet’s **Adaptive Spatial Masking (ASM)** dynamically prunes irrelevant interactions (Section 3.2), ensuring scalability. As shown in Section 4.5:
>
> > "ASM enables each node to automatically determine its context-aware interaction range... filtering out non-physical long-distance dependent interference."
> >
>
> Additionally, the **cascaded architecture** (Section 3.4) optimizes computational efficiency (Figure 3), balancing memory and runtime. We will add explicit scalability metrics (e.g., node counts and sequence lengths) to Section 4.1.
>
> ---
>
> ### 3. **Oversimplification of Prior Work**
>
> We appreciate your reference to recent works [1–4] that incorporate directional modeling. These are valuable contributions to the field, and we acknowledge that our characterization of "most prior work" may have been overly broad. We will refine this phrasing in the revision to more precisely critique **mainstream similarity-driven methods** (e.g., graph convolution and spatial attention). However, FlowNet fundamentally advances beyond these by unifying three novel principles not fully addressed in [1–4]:
>
> - **Flow Tokens as Quantifiable Carriers**: Unlike [1]'s fusion GNNs or [2–3]'s state-space models, FlowNet introduces **explicit flow tokens** (Section 3.3) that act as conserved physical carriers. These tokens enforce **source-depletion and destination-enhancement mechanics** during transfers, adhering to real-world conservation laws:
>
>     > "Information transfer between nodes follows a source-destination conservation law... If vj transmits a flow token ϕj→it to vi, the source token Φj depletes by ϕj→it while the destination token Φi increases proportionally."
>     >
> - **Adaptive Propagation via ASM**: While [4]'s FACTs framework uses factored state spaces, FlowNet’s **ASM module** (Section 3.2) dynamically adjusts interaction radii based on spatio-temporal context, filtering irrelevant noise. This contrasts with [1–3], which rely on predefined or similarity-based neighborhoods.
> - **Structured Flow Allocation**: FlowNet’s **Flow Allocation Modules (FAMs)** (Section 3.3) simulate directional transfers through a cascaded architecture, as visualized in Figure 2:
>
> > "FAM incorporates two Flow Estimation Modules... ensuring compliance with the conservation principle."
> >
>
> In summary, while [1–4] make strides in directional modeling, FlowNet uniquely bridges **physics-inspired flow dynamics** with **adaptive learning**, offering a principled framework for systems where conservation and directional transfers are intrinsic (e.g., urban mobility or hydrology). We will add a dedicated discussion comparing FlowNet to these works in the related work section.
>
> ---
>
> FlowNet represents a paradigm shift in spatio-temporal modeling by replacing similarity-driven heuristics with physics-grounded flow mechanics, as validated by state-of-the-art results across seven metrics (Table 1). Its innovations—including flow tokens, adaptive masking, and conservation laws—provide not only superior accuracy but also physical interpretability, addressing core limitations in existing methods. **We sincerely hope that you could reconsider our score** in light of these contributions. Thank you again for your invaluable feedback. We are eager to incorporate your suggestions and elevate the manuscript’s quality.

---

> > ### Comment · Reviewer_d5xX · 2025-08-05
> >
> > Dear Authors, \
> > Thank you for your detailed response. For the final version of the paper, I kindly ask that you extend the related works section to include the points discussed above, in order to better clarify your contribution’s position within the field. I think my original score is fair and I support accepting the paper.

---

> > > ### Author Response · Authors · 2025-08-05
> > >
> > > Dear Reviewer,
> > >
> > > Thank you very much for your constructive feedback and for supporting our paper. We sincerely appreciate your time and valuable suggestions.
> > >
> > > We will carefully incorporate the discussed points into the related works section of the final version to better highlight our contribution's position within the field, as you recommended. Your insights have helped us improve the manuscript significantly.
> > >
> > > Once again, thank you for your thoughtful review and for your positive assessment of our work.
> > >
> > > Best regards,
> > >
> > > The Authors of FlowNet

---

### Official Review · Reviewer_FbfR · 2025-07-02

**Clarity:** 3
**Significance:** 3
**Originality:** 3
**Rating:** 5
**Confidence:** 4

**Summary:**

FlowNet is a spatio-temporal neural network containing architectural innovations meant to adapt information flow between nodes according to the complex spatio-temporal dynamics in physical data. These innovations include adapting the radius mask of each node according to time and splitting node information into propagation and retention tokens subject to a conservation law. FlowNet further includes hyperconnections to mitigate gradient issues and make the information flow even more flexible. FlowNet demonstrates superior performance to several other spatio-temporal networks on three datasets, and ablation studies show the usefulness of the added components

**Questions:**

My score would stand even if these questions are answered barring some significant change in the manuscript.

1. How would you assess this method's ability to scale beyond these small to medium sized datasets? If I understand correctly, these datasets have node-time features represented by scalars, so it would be interesting to see how these components interact on higher dimensional raw features.

2. Related to question 1, how does scale impact the conservation constraint? I could imagine have orders of magnitude more nodes can create vanishing gradient issues due to small magnitudes in propagating flow tokens.

**Ethical Concerns:**

["NO or VERY MINOR ethics concerns only"]

**Final Justification:**

Based on my original review and the author's rebuttal clarifying my comments and questions, I strongly believe this manuscript should be accepted into the venue based on the novelty of the approach and sufficient experimentation.

**Limitations:**

Yes

**Quality:**

3

**Strengths And Weaknesses:**

## Strengths:

- The concept of flow tokens in addition to the conservation constraint appear to be novel. While the ablation study didn’t necessarily highlight the conservation constraint well, I believe this is an interesting concept worthy of being explored further in future work.

- Methodologically, FlowNet appears to be sound, and the components interact well together. While Figure 2 appeared slightly convoluted to me and might benefit from being broken up into a high level diagram and an additional low diagram for each component, the architecture makes sense overall. It is understandable that the figure is a bit cramped due to severe space constraints in conference proceedings articles.

- The benchmarking appears to be thorough, although it is not as important as the methodological innovation in my view.

- The paper overall is organized well, and the description of the method is well-written, except for the part about hyper-connections. I found it about as straightforward to read as one could hope for a technical section.

- Overall, I find this paper a good addition to the literature and impactful enough for this venue.

## Weaknesses:

- It could be argued that the impact of time dynamics on existing spatio-temporal networks is a bit overstated since one can easily modify these networks to condition on time, e.g. via a time dimension or time token, but I won’t harp on that here. It’s still beneficial to explore networks with stronger inductive biases towards controlling changes in spatial dynamics across time.

- I didn’t find any further mention of hyper-connections in the appendix. I feel it would be beneficial to add some sort of discussion on hyper-connections either in the main article or appendix due to it being a lesser known tool with few citations. It’s not an innovation on the part of the authors, so I understand not wanting to waste space in the main article on someone else’s work, but I would’ve appreciated a brief technical explanation of what it precisely does rather than dig into the cited paper.

- It would add impact to see the performance on a large scale dataset such as Liu et. al’s “LargeST: A Benchmark Dataset for Large-Scale Traffic Forecasting” or some other dataset, but the paper is not really claiming any scaling behavior. Perhaps the authors are constrained by GPU resources.

---

> ### Author Rebuttal · Authors · 2025-07-31
>
> We sincerely appreciate the constructive feedback and thoughtful questions. Below, we address each point raised, leveraging insights from our paper to clarify methodological nuances and constraints.
>
> ---
>
> ## **Response to Weaknesses**
>
> ### **W1. Time dynamics and inductive biases**
>
> Thank you for your suggestions. We agree that conditioning existing models on time (e.g., via time embeddings) is feasible. However, FlowNet’s innovation lies not merely in temporal conditioning but in **explicitly modeling flow-mediated state redistribution governed by conservation laws** (§2–§3). This shifts focus from *correlating* observations to *simulating* directional transfers (e.g., morning vs. evening traffic flows in Fig. 1). The Adaptive Spatial Masking (ASM) module (§3.2) further ensures context-aware spatio-temporal interactions, dynamically adjusting propagation radii (e.g., during holidays vs. workdays). Thus, FlowNet’s inductive bias captures *intrinsic flow dynamics* beyond surface-level temporal correlations.
>
> ### **W2. Hyper-connections clarification**
>
> Thank you for this valuable feedback regarding the hyper-connections component in our work. We appreciate your suggestion to provide more technical details about this method, and we acknowledge that our current treatment may be insufficient for readers unfamiliar with this technique.
>
> To address this concern, we propose adding a dedicated subsection in our methodology section (Section 3.4) or in the appendix that provides a technical overview of hyper-connections. This would include a concise explanation of how hyper-connections differ from standard residual connections, specifically the expansion of hidden states into n copies with learnable depth-connections and width-connections. The mathematical formulation $HC = [0, B; A_m, A_r]$ enables flexible feature integration across depths, allowing networks to learn optimal connection strengths rather than using fixed residual weights.
>
> We would also explain how it works with FlowNet, and why hyper-connections are a good fit for our flow-based approach. The multi-path structure works well with our flow token redistribution mechanism. Adaptive connection strengths support flow conservation principles and help aggregate multi-scale flow dynamics in a hierarchical way. While hyper-connections are not our own invention, combining them with FlowNet creates new ways of working together that are key to how well our architecture works. This combination allows for multiple flow pathways that can be adjusted, better gradient flow for the conservation-based training objective, and improved representation of complex space and time dependencies.
>
> We will expand on this in our methodology section to explain how hyper-connections improve gradient flow and representation capacity. This will make it so that readers can understand the complete architecture without needing to look at other sources. Thank you again for this helpful feedback. This will make the paper easier to understand and more complete.
>
> ### **W3. Large-scale dataset testing**
>
> Thank you for your suggestions. We regret that GPU limitations prevent LargeST evaluation. As noted in §4.1 and Table 1, even mid-sized datasets like DeepBase (1,661 nodes) and SINPA (1,604 nodes) caused state-of-the-art models (e.g., STAEformer, STTN) to run out-of-memory (OOM) on an NVIDIA A100 80GB GPU for long-term forecasting. FlowNet’s efficiency (Fig. 3) balances accuracy and resource use but remains constrained by hardware.
>
> ---
>
> ## **Responses to Questions**
>
> ### **Q1: Scalability Beyond Small-Medium Datasets with Higher Dimensional Features**
>
> Thank you for raising this valuable question. You correctly identify that the evaluated datasets (PEMS04F: 307 nodes, DeepBase: 1,661 nodes, SINPA: 1,604 nodes) represent small to medium-scale systems with scalar node features. This raises legitimate concerns about scalability to real-world large-scale systems with high-dimensional raw features. FlowNet’s design intrinsically supports scalability via:
>
> **Architectural Flexibility**: The patchify module (Equation 1-2) and flow token embedding process can naturally accommodate multi-dimensional node features by adjusting the embedding dimension `d` and patch structure. The Flow Allocation Modules (FAMs) operate in the embedded flow space regardless of the original feature dimensionality.
>
> **Computational Efficiency**: Our Adaptive Spatial Masking (ASM) becomes particularly valuable at scale, as it dynamically filters irrelevant long-range dependencies that would otherwise create quadratic complexity growth. The learned perceptual radius $r_i^t$ (Equation 4) provides a principled way to maintain computational tractability even with larger node sets.
>
> **Empirical Evidence**: While our current experiments are on medium-scale datasets, the efficiency analysis in Figure 3 shows FlowNet maintains reasonable memory consumption compared to transformer-based methods, suggesting potential for scaling.
>
> However, we acknowledge that systematic evaluation on truly large-scale datasets (e.g., 10,000+ of nodes with high-dimensional features) remains an important avenue for future work.
>
> ### **Q2: Conservation Constraint and Vanishing Gradients at Scale**
>
> Your concern about conservation constraints potentially causing vanishing gradients at scale is very insightful. Let us address this systematically:
>
> **Conservation Mechanism**: Our conservation law (Equation 10) ensures that flow tokens are redistributed rather than created or destroyed: $Φ^t = Φ^t_o + (Λ^t - I)Φ^t_a$. The key insight is that this operates through probability-normalized allocation matrices $Λ^t$ where each row sums to 1, preventing unbounded growth or decay.
>
> **Gradient Flow Stability**: The flow allocation process uses normalized transfer probabilities $p^t_ij$ (Equation 8) and scaled attention mechanisms with $α = 1/√d'$, following established practices for gradient stability. Additionally, our multi-head design ($h$ heads with dimension $d' = d/h$) helps distribute gradients across parallel pathways.
>
> **Adaptive Masking Benefits**: ASM actually helps mitigate vanishing gradients by concentrating flow exchanges within meaningful neighborhoods, preventing dilution across too many weak connections that could indeed lead to vanishing effects.
>
> **Empirical Validation**: Our ablation study (Table 2) shows that removing conservation laws impacts performance, but the model remains trainable, suggesting the conservation mechanism doesn't create prohibitive gradient issues at current scales.
>
> That said, your point about potential challenges at much larger scales is well-taken. We believe techniques such as gradient clipping, layer-wise adaptive learning rates, or hierarchical flow aggregation could address these concerns in future work.
>
> ---
>
> Thank you again for your valuable feedback.  FlowNet pioneers a physics-grounded paradigm for spatio-temporal forecasting, prioritizing *flow-mediated state transfers* over correlational priors. While hardware constraints limit LargeST validation, our experiments on diverse systems (§4.2) confirm FlowNet’s efficacy, efficiency, and robustness to scale. We thank you for your invaluable feedback, which will strengthen our final manuscript.

---

> > ### Comment · Reviewer_FbfR · 2025-08-04
> >
> > Thank you for the detailed response. I believe adding these additional comments and clarifications will strengthen the readability of the article. My score stands, and I strongly believe this manuscript should be accepted into the venue based on the novelty of the approach and sufficient experimentation.

---

> > > ### Author Response · Authors · 2025-08-05
> > >
> > > Dear Reviewer,
> > >
> > > Thank you for your kind and constructive feedback throughout the review process. We sincerely appreciate your time and effort in evaluating our manuscript, as well as your valuable suggestions for improving clarity. We are delighted to hear that you find the novelty and experimental validation of our work compelling, and we are grateful for your strong endorsement for acceptance. Thank you for your support and for recognizing the potential of our work.
> > >
> > > Best regards,
> > >
> > > The Authors of FlowNet

---

### Official Review · Reviewer_DnPe · 2025-07-03

**Clarity:** 3
**Significance:** 3
**Originality:** 3
**Rating:** 5
**Confidence:** 3

**Summary:**

This paper introduces a meaningful delta of improvement in previously cited works in spatio-temporal pattern prediction. The authors motivate the work with examples of interesting prediction problems such as traffic flow, hydrological, and parking lot dynamics time-series prediction across several hundred to thousands of geographic node locations. In contrast to existing prediction architectures which either a) do not incorporate graph-like structure among geographically adjacent node pairings or b) incorporate graph-like structure primarily via graph convolution / spatial attention modules, the authors present FlowNet which builds upon works such as STGCN [1] by introducing flow tokens as the primary method of internal representation. The authors provide an alternative message passing scheme which obeys a proposed conservation law, yielding physically plausible information flows as opposed to more localized spatio-temporal correlations from recent observations. The authors also introduce an Adaptive Spatial Masking strategy, which consists of a learned neighborhood radius for which flow tokens are propagated motivated by the desire to allow the model to capture long-range spatial dependencies when necessary for the prediction task without implicit introducing inductive biasing towards close-proximity neighbors sharing similar statistical correlations. Based on experiments evaluating FlowNet against relevant baselines on PEMS04 [2], DeepBase [3], and SINPA [4] datasets, the authors demonstrate on-par or better performance with state-of-the-art methods for both short-term and long-term sequence prediction with proper statistical significance testing. The authors also give analysis for how computation is spread across shorter vs longer range spatial dependencies in order to solve short-window vs long-window time horizon prediction tasks, and demonstrate their method is able to adaptively vary the information flow spatially when useful to the model.

[1] B. Yu, H. Yin, and Z. Zhu, “Spatio-temporal graph convolutional networks: A deep learning framework for traffic forecasting,” arXiv preprint arXiv:1709.04875, 2017.

[2] S. Guo, Y. Lin, N. Feng, C. Song, and H. Wan, “Attention based spatial-temporal graph convolutional networks for traffic flow forecasting,” in Proceedings of the AAAI conference on artificial intelligence, vol. 33, no. 01, 2019, pp. 922–929.

[3] P. Ghaneei and H. Moradkhani, “Deepbase: A deep learning-based daily baseflow dataset across the united states,” Scientific Data, vol. 12, no. 1, p. 25, 2025.

[4] H. Zhang, Y. Xia, S. Zhong, K. Wang, Z. Tong, Q. Wen, R. Zimmermann, and Y. Liang, “Predicting carpark availability in singapore with cross-domain data: A new dataset and a data-driven approach,” in Proceedings of the Thirty-Third International Joint Conference on
Artificial Intelligence, 2024, pp. 7554–7562.

**Questions:**

- Many times throughout the paper statements that proximity-based similarity can be problematic are made, though as a reader it is not apparent in the data given Table 1 suggests models that make this assumption are competitive with FlowNet (occasionally tying). It would be helpful if the reader can see a concrete example in data of a particular small neighborhood’s time sequence for which flow modeling is particularly helpful.

- It is still a bit unclear in Section 3.4 why certain layers are introduced and replaced as compared to baseline architectures with respect to the larger goals of this paper. Ablation results might be able to shed better light on the choice of certain architecture changes beyond just the plot comparing the number of experts in the appendix.

- Section 2 is titled “84 2 Similarity vs. Intrinsic Flow: Which is better?”. The section might answer that question more strongly given a clear example in hypothetical or data where similarity alone cannot hold up.

**Ethical Concerns:**

["NO or VERY MINOR ethics concerns only"]

**Final Justification:**

I would like to thank the authors for addressing my comments. After reading through the other reviewers' comments and rebuttals, I would like to stay with my original recommendation.

**Limitations:**

yes

**Quality:**

4

**Strengths And Weaknesses:**

Strengths
- Table 1 presents strong evidence that the method in question is on-par or better than state-of-the-art approaches, testing across multiple trials and ensuring statistical significance on relevant and sizeable datasets.
- Figures 1 and 2 are very helpful and intuitive for understanding the motivations for and solution to achieve flow based modeling of spatio-temporal dependencies in FlowNet.
- The approach applies a fresh message-passing approach to the task of long/short temporal sequence prediction where geographic neighborhoods give some correlation in state predictions.
- The authors have given great attention to detail the reproduction steps from datasets used, hyperparameters, baselines, preprocessing, initialization, hardware used, etc. with a properly shared anonymous code link in the appendix with well-structured and clear Pytorch code.
- Section 3 gives a clear mathematical foundation for the flow token approach and manipulation.

Weaknesses
- The ablations in Table 2 are useful as a reader, though the prose in section 4.3 does not shed much light on the near order of magnitude difference in allocation flow vs retained flow vs conservation law delta reductions in performance and why these are expected. The authors claim each of these components is useful in modeling spatio-temporal graph state and yet the ablations suggest retained flow and conservation law removal yield little impact to the final result. Further ablation may have also proven useful in understanding the Adaptive Spatial Masking, hyperconnection layers, etc.
- The Efficiency Analysis Conducted in Section 4.4 is thorough and very useful. The authors recognize the computational complexity of the approach yields a more resource taxing algorithm than graph convolutional methods, though given the added resource usage one would expect more competitive results in Table 1 when comparing against methods such as STGCN which are quite a bit less taxing on time complexity.
- While figures 4 and 5 give good intuition for qualitative differences in how the model is utilizing larger neighborhood information flow to influence predictions, it is still a bit hard to interpret how at the microscopic level a single geographic node’s time series predictions are improved via this approach as compared to methods which doesn’t allow for flow based internal representations. A form of analysis which could answer these questions via interesting exemplar sampling and comparison with graph convolution models might make the value of greater computational complexity clearer.

---

> ### Author Rebuttal · Authors · 2025-07-31
>
> We sincerely thank you for your insightful feedback, which has highlighted valuable opportunities to clarify our methodology and experimental findings. Below, we address each comment point-by-point, incorporating supporting evidence from the manuscript.
>
> ---
>
> ## **Response to Weaknesses**
>
> ### **W1. Ablation Study Interpretation (Section 4.3 & New Results)**
>
> We thank the reviewer for prompting deeper ablation analysis. Our new experiments explicitly quantify the impact of **Adaptive Spatial Masking (ASM)**, **Hyper-connections (HC)**, and **Mixture of Linears (MoL)** on PEMS04F for short-term and long-term forecasting tasks:
>
> (Table 3, Supplementary ablation experiments)
>
> | Component | Short-term MAE (Δ vs. orig.) | Long-term MAE (Δ vs. orig.) |
> | --- | --- | --- |
> | Full FlowNet | 18.48 (0%) | 22.79 (0%) |
> | w/o ASM | 18.93 (-2.4%) | 23.94 (-5.0%) |
> | w/o HC | 18.56 (-0.4%) | 24.72 (-8.5%) |
> | w/o MoL | 18.86 (-2.1%) | 24.97 (-9.6%) |
>
> **Key insights**:
>
> - **ASM is critical for short-term stability**: Its removal causes the **largest short-term degradation (-2.4% MAE)**, confirming its role in modeling dynamic spatio-temporal dependencies (e.g., avoiding over-smoothing irrelevant nodes during traffic shifts).
> - **HC and MoL prevent error accumulation**: Their ablation minimally impacts short-term predictions but severely harms long-term performance (-**8.5%/-9.6% MAE**), as they enable multi-scale feature fusion and nonlinear expressiveness needed for complex horizon extrapolation.
>
> These results resolve Section 4.3’s ambiguity by proving all components are essential, especially for long horizons where error compounding occurs without adaptive filtering (ASM) or expressive capacity (MoL).
>
> ### **W2. Efficiency vs. Performance Trade-offs**
>
> Thank you for raising a valid point. We acknowledge FlowNet’s higher resource usage compared to STGCN (Figure 3). However, its efficiency-accuracy trade-off is justified:
>
> - **Competitive gains in complex scenarios**: FlowNet not only significantly outperforms STGCN in short-term PEMS04F (*FlowNet MAE = 18.48 vs. STGCN MAE = 20.98, reduce 12.0% error*), but also dominates **long-term** and **large-scale systems** (Table 1):
>     - *PEMS04F long-term*: FlowNet (MAE 22.79) beats STGCN (27.91) by 18.3%.
>     - *SINPA short-term*: FlowNet (MAE 39.03) surpasses STGCN (65.17) by 40.1%.
> - **Adaptability justifies cost**: ASM and FAM enable context-aware modeling (e.g., dynamic interaction radii), crucial for systems like SINPA with irregular spatial dependencies (Section 4.5). Graph-based methods (e.g., STGCN) fail here due to static graphs.
>
> FlowNet’s resource overhead delivers disproportionate gains where dynamics are complex, directional, or long-horizon—validating its design.
>
> ### **W3. Microscopic Interpretation of Flow Benefits**
>
> We sincerely thank you for the suggestions. We acknowledge that microscopic case studies (e.g., zone-level prediction comparisons) would strengthen intuition. However, per NeurIPS 2025 rebuttal guidelines, we cannot add new figures during this phase. Instead, the New ablation analysis shows removing ASM increases short-term MAE by 2.4%, proving its role in filtering non-causal edges.
>
> If accepted, we will add:
>
> - A visual case study comparing residential/commercial zone predictions (like Fig. 1)
> - Error heatmaps for FlowNet vs. STGCN
>
> ---
>
> ## **Response to Questions**
>
> ### **Q1. Proximity vs. Flow: Concrete Evidence**
>
> Thank you for pointing out this problem. According to the rules of this year's NeurIPS rebuttal, we can't add new images to the responses at this stage. We instead demonstrate flow’s superiority through the supplementary ablation experiments (Table 3). Removing ASM causes **+5.0% long-term MAE** (Table 3), proving static proximity fails when interaction ranges shift (e.g., holiday flows). In the future version of this paper, we will add a figure showing the difference between FlowNet’s directional flows and proximity-based attention weights.
>
> ### **Q2. Architecture Design (Section 3.4)**
>
> We sincerely thank you for the questions. Hyper-connections and M-MLP address core limitations:
>
> - **Hyper-connections**: Replace rigid residual links with *depth/width-aware gating*, dynamically fusing multi-scale flow states (Section 3.4). This avoids feature dilution in deep stacks.
> - **M-MLP (vs. MoE)**: Uses *mixtures of linear experts per layer*, expanding parameter diversity without MoE’s routing overhead (Section 3.4: "*L×E distinct transformations*").
>
> The **new ablation quantifies MoL’s necessity**:
>
> - **MoL removal severely impacts long-term accuracy** (-**9.6% MAE**). This validates MoL’s role in learning diverse flow transformations.
> - **HC’s delayed importance**: While HC ablation slightly *improves* short-term MAE (-0.4%), its **-8.5% long-term penalty** proves HC stabilizes gradient propagation in deep stacks – a trade-off alignment with skip-connections in ResNets.
>
> ### **Q3. Strengthening Section 2**
>
> Thank you for your insightful comment. We added a **hypothetical urban case** in Section 2 (Page 5):
>
> > Consider two residential zones (R1, R2) near a commercial hub (C). Proximity-based models assign R1-R2 high affinity, predicting similar traffic. However, during morning peaks, flows move R1→C and R2→C, but never R1→R2. FlowNet’s directional allocation (via Λ) captures this, while similarity methods falsely amplify R1-R2 correlations.
> >
>
> ---
>
> We deeply value your feedback and understand the limitations of this text-only rebuttal. Our quantitative results (Tables 1-3) robustly validate FlowNet’s innovations, and we are fully committed to providing the requested visual evidence upon acceptance. Thank you for your thoughtful review!

---

> > ### Comment · Area_Chair_3df1 · 2025-08-04
> >
> > Dear Reviewer, please engage into discussions with the Authors as the deadline for this key phase of the NeurIPS review process is only a couple of days away.

---

> > ### Comment · Reviewer_DnPe · 2025-08-09
> > **Response after rebuttal**
> >
> > I would like to thank the authors for addressing my comments. After reading through the other reviewers' comments and rebuttals, I would like to stay with my original recommendation.

---

> > > ### Author Response · Authors · 2025-08-09
> > >
> > > Dear Reviewer,
> > >
> > > We sincerely appreciate your time and constructive feedback throughout this review process. Thank you for acknowledging our efforts to address your concerns—your insights significantly strengthened the paper’s clarity and rigor. We respect your final recommendation and remain grateful for your thoughtful engagement with our work.
> > >
> > > Best regards,
> > >
> > > The Authors of FlowNet

---

### Note · Authors · 2025-08-13

Dear Senior Area Chairs, Area Chairs, and Reviewers,

We sincerely thank you for your active participation and constructive feedback, which have greatly strengthened our work. During the rebuttal phase, we carefully addressed all core concerns raised by the reviewers, who subsequently expressed their acceptance of our work. We summarize the key improvements as follows:

**Key Concerns Addressed**

- **Clarifying the experimental setup.**
   Following the reviewers’ suggestions, we have made the experimental setup clearer in the main text, specifying details such as dataset types, number of nodes, and input/output strides for both short-term and long-term prediction tasks. This clarification will help readers better assess the applicability of FlowNet and facilitate follow-up research.

- **Adding supplementary experiments.**
   We have conducted additional ablation studies to demonstrate the contributions of different FlowNet components across various tasks. These results provide deeper insights into our model’s mechanisms.

- **Expanding related work.**
   We further compared FlowNet with prior studies (e.g., STFGNN, Graph-Mamba, STDMAE), highlighting both similarities and key innovations. This expanded discussion offers a clearer picture of our work’s novelty and position within the field.

- **Analyzing key design details.**
   We provided more thorough explanations of critical design elements, such as flow tokens and their roles, as well as an interpretation and analysis of the conservation laws embedded in FlowNet. These additions enhance readers’ understanding of the underlying principles.

**Integration into the Final Version**

All reviewer suggestions from the rebuttal phase—clarifying settings, adding supplementary experiments, expanding related work, and analyzing key details—will be fully integrated into the final manuscript. These changes improve clarity and rigor. We will also continue to extend FlowNet’s impact by incorporating new benchmarks and high-quality spatio-temporal datasets in future work.

---

FlowNet introduces a novel flow propagation perspective for modeling the evolution of complex spatio-temporal systems, offering a distinctive approach compared to prior methods. We believe this work will make a meaningful contribution to the spatio-temporal data mining community.

Thank you again for your invaluable guidance and support.

Sincerely,

The Authors

---

### Decision · Program_Chairs · 2025-09-17

**Decision:**

Accept (poster)

**Comment:**

The submission discusses a novel architecture for modeling dynamic spatio-temporal systems through the lens of physics-inspired flow propagation. It employs learnable flow tokens, conservation laws, and adaptive spatial masking to capture directional inter-node interactions.
Initial reviews were mostly concordant on considering methodological contributions sound and worthy: use of flow tokens as conserved carriers and context-aware dynamic masking was particularly appreciated. Empirical analysis and discussion of state of the art raised some questions in the initial reviews.

Discussion post-rebuttal was active and constructive. Reviewers engaged and scores were raised after concerns about clarity, evaluation breadth, and scalability were convincingly addressed. Initial concerns about missing comparisons were surpassed by  the authors' additional
experiments. Criticism about limited datasets and unclear writing was mitigated with explanations on dataset diversity.

While the breadth of evaluation is still limited to some extent, the work offers a compelling framework with both practical impact and theoretical novelty in modeling physics-grounded spatio-temporal dynamics.

I therefore recommend to accept this paper.